# Virus glycoprotein nanodisc platform for vaccine analytics

Kimmo Rantalainen [1,2,3], Alessia Liguori[1,2,3], Gabriel Ozorowski [3,4], Claudia Flynn[1,2,3], Jon M. Steichen [1,2,3], Olivia M. Swanson [1,2,3,4], Patrick J. Madden [3,5], Sabyasachi Baboo [4], Swastik Phulera [4], Anant Gharpure[4], Danny Lu [1,2,3], Oleksandr Kalyuzhniy [1,2,3], Patrick Skog[1,2,3], Sierra Terada [1,2,3], Monolina Shil[3,5], Jolene K. Diedrich[4], Erik Georgeson [1,2,3], Ryan Tingle[1,2,3], Saman Eskandarzadeh[1,2,3], Wen-Hsin Lee [4], Nushin Alavi [1,2,3], Diana Goodwin[1,2,3], Michael Kubitz[1,2,3], Sonya Amirzehni[1,2,3], Sunny Himansu [6], Devin Sok[1,2,3], Jeong Hyun Lee[1,2,3], John R. Yates III [4], James C. Paulson [1,3], Shane Crotty [3,5,7], Torben Schiffner[1,2,3,8] ✉, Andrew B. Ward [3,4] ✉ & William R. Schief [1,2,3,6] ✉

Transmembrane glycoproteins of enveloped viruses are targets of neutralizing antibodies and essential vaccine antigens. mRNA-LNP technology allows in vivo expression of transmembrane glycoproteins, but in vitro biophysical characterization of transmembrane antigens and analysis of post-immunization antibody responses typically rely on soluble proteins. Here, we present a platform for assembling transmembrane glycoprotein vaccine candidates into lipid nanodiscs. We demonstrate the utility of nanodiscs in HIV membrane proximal external region (MPER)-targeting vaccine development by binding assays using surface plasmon resonance (SPR), ex vivo B cell sorting with fluorescence-activated cell sorting (FACS), and by determining the structure of a prototypical HIV MPER-targeting immunogen nanodisc in complex with three broadly neutralizing antibodies (bnAbs), including MPER bnAb 10E8, to 3.5 Å by cryogenic electron microscopy (cryo-EM), providing a template for structure-based immunogen design. To demonstrate general applicability we characterize Ebola virus glycoprotein nanodiscs. Overall, the platform offers a tool for accelerating development of next-generation vaccines.

In the past two decades, nanodiscs have emerged as a means to encapsulate transmembrane proteins into a stable and native-like environment. In most cases, proteins are extracted from the membrane by detergent solubilization and then assembled into a nanodisc by removal of the detergent in the presence of lipid molecules and a scaffold protein. Since the first apolipoprotein-derived membrane scaffold proteins (MSPs) were introduced, several variations and alternative scaffolds have been described, allowing for a variety of disc sizes and versatile experimental approaches[1–5]. Structural studies of membrane proteins have perhaps benefitted the most from nanodiscs,

[1]Department of Immunology and Microbiology, The Scripps Research Institute, La Jolla, CA, USA. [2]IAVI Neutralizing Antibody Center, The Scripps Research Institute, La Jolla, CA, USA. [3]Consortium for HIV/AIDS Vaccine Development (CHAVD), The Scripps Research Institute, La Jolla, CA, USA. [4]Department of Integrative Structural and Computational Biology, The Scripps Research Institute, La Jolla, CA, USA. [5]Center for Vaccine Innovation, La Jolla Institute for Immunology, La Jolla, CA, USA. [6]Moderna Inc., Cambridge, MA, USA. [7]Department of Medicine, Division of Infectious Diseases and Global Public Health, University of California, San Diego (UCSD), La Jolla, CA, USA. [8]Institute for Drug Discovery, Leipzig University Medical Faculty, Leipzig, Germany. ✉e-mail: torbens@scripps.edu; andrew@scripps.edu; schief@scripps.edu

resulting in important advances in understanding of membrane protein structure-function relationship[6,7]. They have also been used successfully in a variety of functional and biophysical approaches, such as surface plasmon resonance (SPR)[8–10].

Viral glycoproteins of enveloped viruses are membrane proteins and essential targets for vaccine development. Structure-based vaccine design was an integral part of the successful COVID-19 vaccine development[11–13], and a significant portion of the groundwork stemmed from decades of HIV vaccine development[14]. Structure-based methods, combined with rapidly advancing computational protein design approaches and mRNA lipid nanoparticle (mRNA-LNP) technology, are now revolutionizing iterative vaccine development[15,16]. Immunogen delivery by mRNA-LNP expands the available immunogens to more native-like constructs that may contain the membrane-proximal epitopes, such as the recently developed cleavage-independent NFL trimers expressing HIV Env[17] and the MD39.3 gp151 Env trimer used in the HVTN302 clinical trial (NCT05217641), but recombinant transmembrane glycoprotein production for biophysical characterization of the immunogens remains challenging. In most studies, membrane proteins are truncated before the transmembrane domain to obtain soluble ectodomains with higher expression levels and easier handling and applicability in downstream analytical methods. The regions excluded typically include the membrane-proximal external region (MPER), transmembrane domain (TM), and intracellular C-terminal domain (CT). In influenza HA, the anchor region analogous to HIV MPER joins the ectodomain as flexible linkers to three transmembrane helices and allows the trimer to tilt in relation to the bilayer surface[18]. Both Env MPER and HA anchor have highly conserved epitopes for neutralizing antibodies (nAbs), making them attractive targets for vaccine development[19–23]. Significant progress in MPER-targeted HIV vaccine development has been made recently with peptide-liposome formulations showing induction of heterologous neutralizing antibody B cell lineages in humans, and with mRNA delivered germline-targeting (GT) epitope scaffold efficiently inducing antibody precursors in non-human primates (NHPs) and mice[24,25]. Analysis of antibodies elicited by these vaccines, as well as further immunogen development, would benefit from inclusion of the full epitope in recombinantly expressed proteins. For example, recombinant proteins matching the transmembrane immunogens delivered as mRNA are required for accurate immunogen assessment in iterative vaccine design methods, such as SPR and structural studies. Importantly, transmembrane versions of Env may also be required for more accurate representations of the glycan shield, as many neutralizing antibodies bind specifically to glycans, and glycoproteins expressed as transmembrane proteins have been shown to have glycan shield compositions closer to the native glycan shield[26–28].

Some uses of nanodiscs to support vaccine development have been demonstrated, particularly in efforts to characterize the structures of HIV MPER-targeting HIV antibody epitopes and transmembrane domains of virus glycoproteins in their native membrane bilayer environment[29–33]. The nanodisc assembly platform presented here provides the reproducibility, scalability, and accurate replication of the vaccine candidate properties that enable routine use of transmembrane viral glycoproteins in key methods employed in iterative, rational vaccine design. We demonstrate the use of nanodiscs to measure antibody affinities by SPR, as FACS probes to sort antigen-specific B cells from mouse and non-human primate (NHP) immunization experiments, and for determining the cryo-EM structure of a prototype HIV boost vaccine candidate in complex with three broadly neutralizing antibodies (bnAbs) that target distinct sites of vulnerability on the HIV Env surface. We highlight the benefits of the approach by reporting the structure of the entire protein epitope of an HIV MPER-targeting bnAb 10E8. Finally, we show the adaptability of the platform to other glycoproteins by assembling engineered Ebola virus glycoprotein in nanodiscs using identical conditions to HIV Env.

## Results

### Glycoprotein nanodisc assembly and analysis workflow

We established a workflow that provides material suitable for methods commonly used in rational, iterative vaccine design (Fig. 1). The workflow was set up using the engineered and highly expressing HIV Env construct BG505 MD39.3 gp151[34,35] that was recently tested in nonhuman primates and a phase 1 clinical trial HVTN 302 (NCT05217641)[36,37]. A linker, followed by an HRV3C protease cleavage site and a strep tag, was added to the intracellular C-terminus, leading to a nanodisc assembly construct base that will hereon be referred to as Env gp151 ND (Fig. 2a, b). This construct was then used to optimize the throughput and reproducibility of nanodisc production, leading to a standardized workflow (Fig. 2c). In short, transmembrane glycoproteins (GPs) were expressed on the surface of human FreeStyle 293-F cells, followed by detergent solubilization with TX-100-containing buffer. As the transmembrane domain of viral glycoproteins is composed of only three helices—one per protomer—detergent solubilization by TX-100 was not expected to disturb the quaternary structure of the engineered GP trimer, as indicated by the binding of structure-specific antibodies to extracted GPs[27,28]. The detergent-solubilized glycoprotein was then assembled into nanodiscs using the MSP1D1 scaffold with or without a biotin tag. While a variety of scaffold proteins are available, we opted to reduce variables and focused on using the standard MSP1D1 scaffold for all purposes. The developed "batch assembly" approach, resembling previously described "on-column"[38] and "on-bead"[39] methods, facilitated reproducibility and efficient assembly while GPs remained bound to the matrix. Empty nanodiscs for control experiments were produced by omitting GP in the assembly step and using the scaffold protein His-tag with NiNTA affinity purification prior to SEC. Utilizing this workflow, up to 12 samples were routinely processed simultaneously in 5 days from transfection to purified nanodisc. One liter of cells yielded approximately 100 to 700 μg of pure GP nanodiscs, which was sufficient for one to three endpoint assays. For example, a single production batch produced EBOV GP nanodiscs, an HIV Env nanodisc-Fab complex for cryo-EM structural studies, and eight FACS probes for B cell sorting. Final GP nanodisc preparations were stable for at least three months at +4 °C based on absorbance at 280 nm, presence of trimeric GP in nanodisc in negative stain EM (ns-EM) 2D class averages, and activity in SPR analysis (Fig. 3b).

The workflow was then piloted in an immunogen development project aiming to increase the affinity of the MPER-targeting antibody 10E8 and therefore guide immune responses to the MPER domain. In the native membrane context, the C-terminal MPER helix that is targeted by the heavy chain complementarity-determining region 3 (HCDR3) of 10E8 is sterically occluded (Fig. 2b), and one of the design goals was to improve the MPER access. To this end, N-linked glycosylation sites at positions N88, N618, and N625 (HxB2 numbering) adjacent to the epitope were removed at the base of the trimer. An additional R696S mutation that improved 10E8 binding while maintaining trimeric Env conformation was discovered by mammalian surface display library screening. Earlier structural analysis indicated that R696 resides at the intersection of the three transmembrane helices[29,40]; therefore, this mutation was predicted to disrupt helix interactions and further open the trimer base. The resulting construct will be referred to as Env gp151 MPER ND. A negative control for MPER antibody binding and a KO FACS probe for epitope-specific B cell sorting, designated as Env gp151 MPER-KO ND, was also generated (Fig. 2a). This construct included MPER mutations W672A, F673R, T676R, N677I, L679M, W680E, Y681W, and K683D, which were previously reported to collectively prevent the binding of MPER-targeting antibodies while preserving the overall trimeric structure of Env and other bnAb epitopes[24], or screened for this study to improve the expression of the construct. The prototypical Ebola GP immunogen EBOV GP Mayinga ND was based on a construct with trimer stabilizing

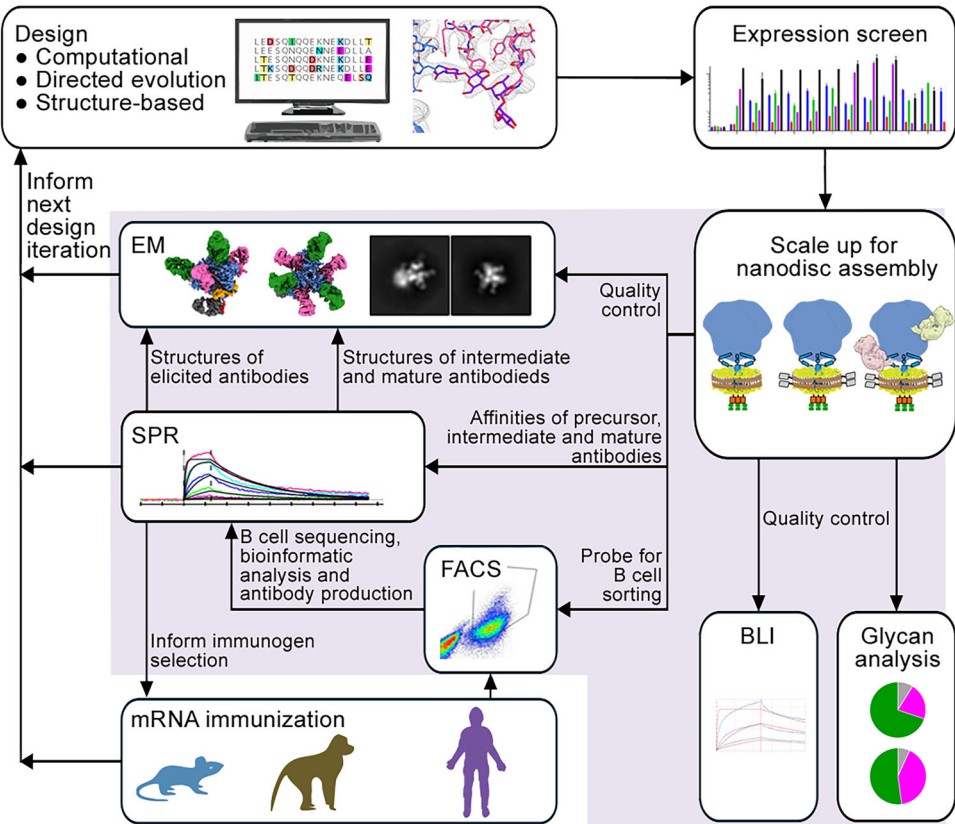

**Fig. 1 | Nanodiscs are used in different steps in iterative rational vaccine design.** Analytical methods that can directly use nanodiscs as a sample material are highlighted.

mutations T577P and K588F[41], to which the native EBOV GP transmembrane (TM) domain and the same intracellular elements as in the HIV Env constructs were added. Sequences of generated constructs are given in Supplementary Table 1. Assembled nanodiscs generally exhibited broad peaks in size exclusion chromatography (SEC) spectra, indicative of extensive glycosylation and variation in hydrodynamic radius that may result from positional heterogeneity of Env in the disc, and >95% purity and bands of the expected molecular weight as examined by SDS-PAGE (Supplementary Fig. 1a, b). Mass spectrometric glycan analysis using the DeGlyPHER method[42] was done prior to and after nanodisc assembly to ascertain site-specific glycan occupancy and processivity of Env constructs. This analysis confirmed the presence of all expected glycans as well as a high proportion of complex glycans in transmembrane constructs characteristic of native, transmembrane HIV Env (Supplementary Fig. 1c)[26–28]. In addition to the MPER-targeting immunogen constructs and EBOV GP Mayinga ND, a germline-targeting HIV vaccine candidate N332-GT5 was produced as a gp151 transmembrane construct and assembled into nanodiscs using the standardized workflow to test general applicability (Fig. 2 and Supplementary Fig. 2a)[35–37,43,44]. Germline-targeting immunogen N332-GT5 gp140, a soluble trimer, showed proportions of complex-type glycans matching closely those of the transmembrane version. All transmembrane Envs showed higher overall glycan occupancy in the gp41 subunit compared to soluble Env gp140.

**GP nanodisc binding kinetics to antibodies measured by SPR**
SPR is an essential method in iterative vaccine design that measures antibody affinities to engineered immunogens, assessing the effect of the designed features as well as affinities of antibodies induced by vaccination (Fig. 1). We set to establish scalable and reproducible conditions for measuring antibody binding kinetics against transmembrane GPs in nanodiscs in three distinct SPR modalities (Fig. 3a).

Modality A utilized an anti-affinity tag capture strategy in which anti-strep-tag antibody was covalently immobilized on the chip surface, glycoprotein nanodiscs were captured using the intracellular strep tag, and Fab analytes were employed to study monovalent interactions. Modality B employed a low surface density IgG capture strategy, in which anti-human IgG capture antibody was first covalently immobilized, and glycoprotein-specific IgG ligands were captured at reduced capture time to limit the ligand density and thereby reduce avidity effects. Glycoprotein nanodiscs were then injected as analytes. In modality C, nanodisc capture was followed by IgG flow as analyte instead of Fabs, allowing a semi-quantitative high-throughput scouting approach with low nanodisc sample consumption and high sensitivity. Establishing these three modalities for nanodisc GPs allowed flexibility in experimental design based on data quality requirements and sample availability.

Next, we used the GP nanodisc SPR modalities to establish baseline kinetics for neutralizing and non-neutralizing monoclonal HIV Env antibodies spanning diverse epitopes (Fig. 3b–d, Supplementary Fig. 2b–d). Modality A measured affinities of 100 to 250 nM for 10E8 to Env gp151 ND design base construct (Fig. 3b) and slightly higher apparent affinity of 36 to 42 nM using modality B. Using modality A, we observed a 70-fold increase in 10E8 affinity (250 nM vs 3.6 nM) to Env gp151 MPER ND as a result of removal of the MPER epitope-proximal glycans and the mutation R696S in TM (Fig. 3c). Incorporation of MPER-KO mutations completely abrogated 10E8 binding. Kinetic analysis indicated that the improved affinity in the engineered immunogen was primarily attributed to a decreased off-rate ($8.1 \times 10^{-3}$ s$^{-1}$ for Env gp151 ND versus $3.5 \times 10^{-4}$ s$^{-1}$ for Env gp151 MPER ND (Fig. 3d). Additional bnAbs targeting the MPER and other epitopes on the surface of Env were also tested for more comprehensive antigenic profiling. This confirmed binding of MPER-targeting antibodies to Env gp151 ND (Supplementary Fig. 2b), and showed as expected that non-

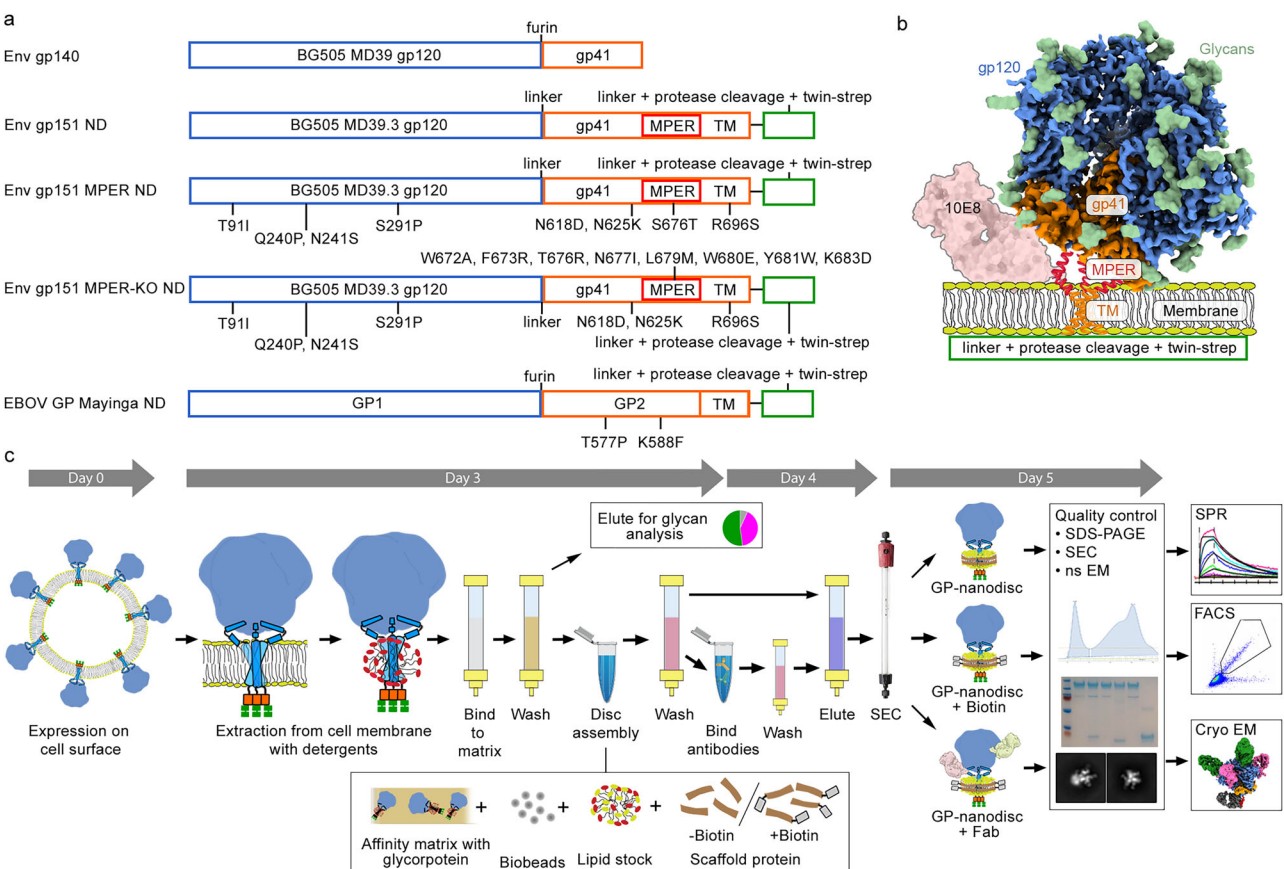

**Fig. 2 | HIV Env constructs used for method development and a general overview of the nanodisc assembly workflow. a** Naming and gene organization of the glycoprotein constructs used for nanodisc assembly. Introduced mutations and intracellular elements are indicated. ND refers to constructs made for the nanodisc platform. **b** Schematic overview of HIV Env constructs and MPER targeting antibody 10E8 binding. **c** Key steps of the 5-day workflow for glycoprotein nanodisc assembly. Glycoprotein was extracted with a detergent from the cell surface and bound to the affinity matrix. Disc assembly was performed "in-batch" while bound to the affinity purification matrix. Assembled discs were then eluted for final SEC polishing purification, followed by quality control steps before being subjected to final analytical methods (SPR, FACS, cryo-EM).

MPER bnAbs have similar affinities for soluble Env gp140 and Env gp151 ND (Supplementary Fig. 2c). Non-neutralizing antibodies (non-nAbs) RM19R and RM20A3 exhibited reduced or no binding to Env gp151 ND as compared to Env gp140 (Supplementary Fig. 2d). These two non-nAbs target the base of the trimer, which is exposed in Env gp140, whereas in Env gp151 ND access is presumably restricted by MPER and TM domains as well as the lipid surface of the nanodisc. Finally, we employed modality C to confirm the binding of two Ebola GP glycan cap-specific antibodies EBOV-296 and 13C6 to EBOV GP nanodiscs, followed by monovalent affinity determination using modality A (Fig. 3e). As expected, antibodies EBOV-296 and 13C6 bound with high affinity (46 nM and 16 nM, respectively) to EBOV GP nanodiscs but did not interact with HIV Env gp151 ND. Conversely, HIV MPER bnAb PGZL1 bound to Env gp151 ND but did not interact with the EBOV GP ND. Collectively, these data demonstrate that GP nanodiscs and SPR can be used to characterize the antigenic landscape of the designed trans-membrane immunogen, inform immunogen selection for in vivo studies, identify suitable complexes for structural studies by cryo-EM, and support GP nanodisc use as B cell sorting probes in FACS (Fig. 1).

### FACS sorting of B cells from immunized animal models using nanodisc GP probes

Single B cell sorting and sequencing are routinely used to analyze B cell responses induced in vivo. Typically, biotin-tagged variants of the immunogen are used with streptavidin. This allows conjugation of the same probe to multiple fluorochromes to increase selection specificity and to prevent isolation of fluorochrome-specific B cells. By including a matching sorting probe with KO mutations in the targeted epitope, epitope-specific responses can be separated from off-target responses against unrelated epitopes. Soluble GP protein constructs solely consisting of the ectodomain are most often used as antigenic baits. For immunization studies where membrane-proximal epitopes are of interest, such as the HIV MPER, soluble probes would not be able to extract all appropriate responses to the targeted epitope due to several challenges which include (1) the hydrophobicity of the epitope regions which make bait production challenging; (2) non-native or incomplete structural conformation of the epitope in shorter peptide representations of the antigen; and (3) missing lipid membrane components that constitute a portion of the native epitope. Here, we used a biotinylated MSP1D1 scaffold to generate nanodisc GP FACS probes. We first tested Env gp151 ND probe binding to FACS compensation beads conjugated to 10E8, DH511, or BG18 bnAbs (Fig. 4a). The background signal was measured as binding to either MSP1D1 scaffold protein alone, empty nanodisc assembled with DOPC lipids, or to empty nanodisc with a mixture of neutral and charged lipids. Env gp151 ND exhibited substantially higher signal compared to controls, confirming that adequate separation from background signal can be detected. We detected no binding with any of the tested probes to unimmunized C57BL/6 (B6) mouse splenocytes, and a strong signal against engineered Ramos cells expressing the bnAb VRC01 as its B cell receptor (BCR; Fig. 4a)[45]. This data indicated that Env nanodisc probes were

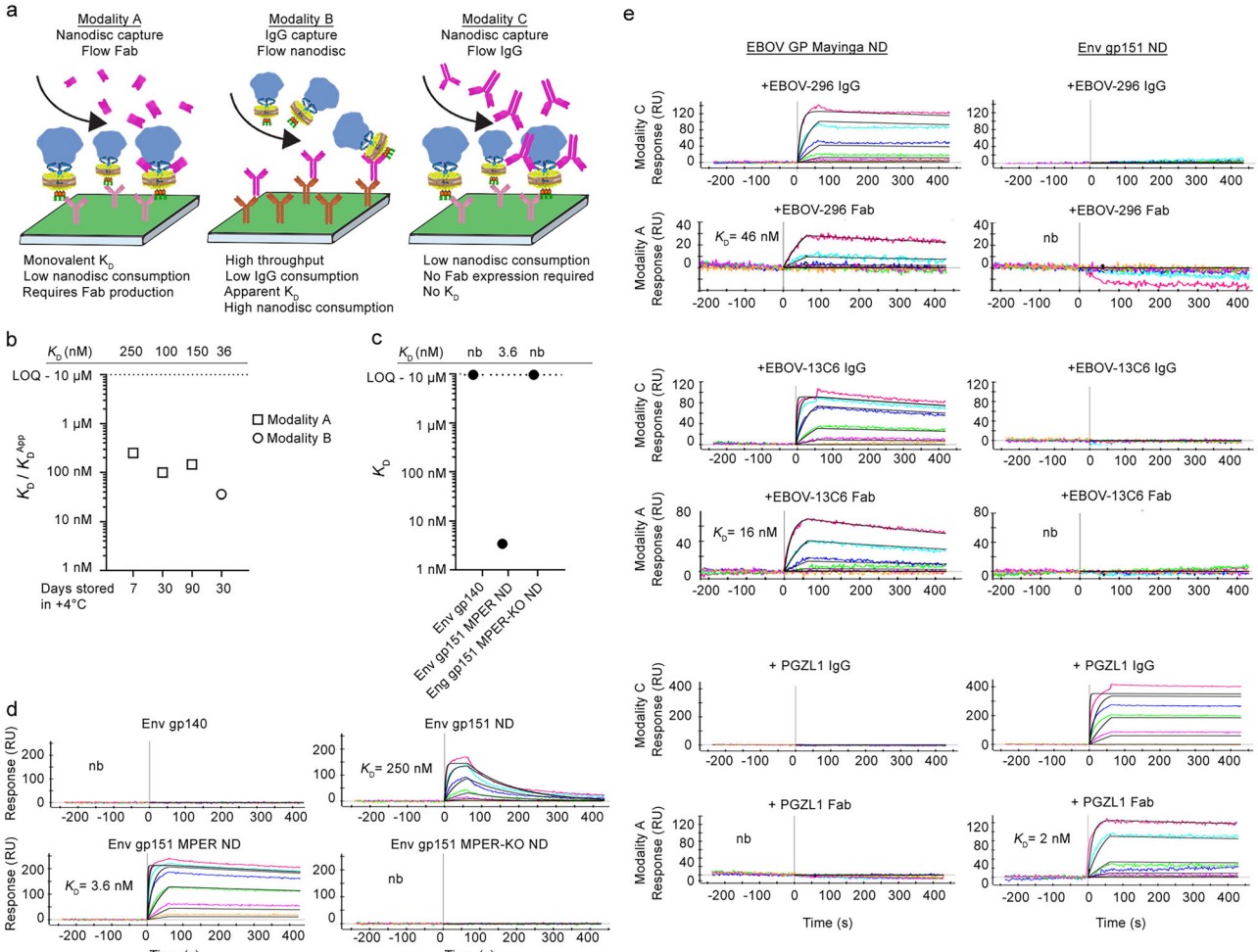

**Fig. 3 | Different SPR modalities and kinetic analyses of GP nanodiscs assembled with DOPC lipids. a** Schematic illustration of three SPR modalities used throughout the study and their strengths and weaknesses. **b** Affinity and apparent affinity of MPER targeting 10E8 antibody to Env gp151 ND as measured with modalities A and B, respectively, after given storage time. **c** Affinity of 10E8 against engineered Env constructs using modality A. **d** Modality A SPR data and affinities presented in b and c, showing an increase in affinity originating from reduced off-rate in Env gp151 MPER ND. **e** Modality C was used to scout Ebola-specific EBOV-296 and EBOV-13C6 IgG epitope accessibility in EBOV GP nanodiscs. The same antibodies were expressed as Fabs for $K_D$ measurements with modality A. Data show binding of two anti-Ebola antibodies targeting the glycan cap (EBOV-296 and 13C6). Env gp151 ND and anti-HIV MPER antibody PGZL1 were used as negative controls. LOQ, limit of quantification. Source data are provided as a Source Data file.

suitable for the detection of B cells from mice by flow cytometry that specifically target Env.

We then proceeded with a pilot immunization experiment in mice to explore if vaccine-elicited antigen-specific B cells could be identified using Env nanodiscs probes. We utilized transgenic mice expressing the human D3-3 and $J_H6$ genes that allow mice to express long HCDR3s, including HCDR3 precursors of HIV MPER bnAbs 10E8, DH511, and LN01[24]. Mice were intramuscularly injected with 10 µg of mRNA-LNPs encoding HIV Env gp151 MPER ND construct. Animals were sacrificed, and spleens and lymph nodes were harvested 6 weeks post-immunization. B cells were stained with immunogen-matched (WT) nanodisc probes on two distinct fluorophores and Env gp151 MPER-KO ND nanodisc probe with KO mutations in the MPER to identify MPER-specific cells (Fig. 4b and Supplementary Fig. 3). Among class-switched memory B cells (defined as CD19+IgD-IgM-), 0.7% of cells were determined to be antigen-specific, compared to 0.15% within the IgD+IgM+ naïve B cell population (Fig. 4c). This ~4.7-fold increase was statistically significant ($p = 0.0177$), indicating that the mRNA immunization elicited HIV Env gp151 MPER-specific responses that could be detected by Env nanodisc probes. The proportion of WT Env nanodisc-binding B cells that selectively bound to

the 10E8 epitope (%KO- of WT++ B cells) was indistinguishable between the naïve and memory B cell compartments, suggesting that the immunogen did not elicit MPER epitope-specific responses, and that further engineering or inclusion of a prior germline-targeting priming immunogen in the immunization regimen would be required to achieve that capacity[24,46].

Antigen-specific (WT++) mouse memory B cells were sorted and their BCRs sequenced to assess the robustness of nanodisc probes in the selection of antigen-specific BCRs. Eight antibodies from each animal were randomly selected for monoclonal antibody production. An additional 12 antibodies with long (≥19 amino acid (aa)) HCDR3s were selected, and duplicate antibodies (containing identical heavy- and light chains) were removed, resulting in 54 total antibodies. Genes encoding these antibodies were synthesized, and 49 of 54 antibodies were successfully expressed, purified and used in SPR modality B to measure affinities against Env nanodiscs (Fig. 4d). At 1 µM analyte concentration, no non-specific responses to empty nanodiscs were detected. Out of the tested antibodies, 47% bound to soluble gp140 matching the immunogen ectodomain, and 65% bound to Env gp151 MPER ND or Env gp151 MPER-KO ND. Considerable binding to the ectodomain, similar binding to the MPER-targeting immunogen and its

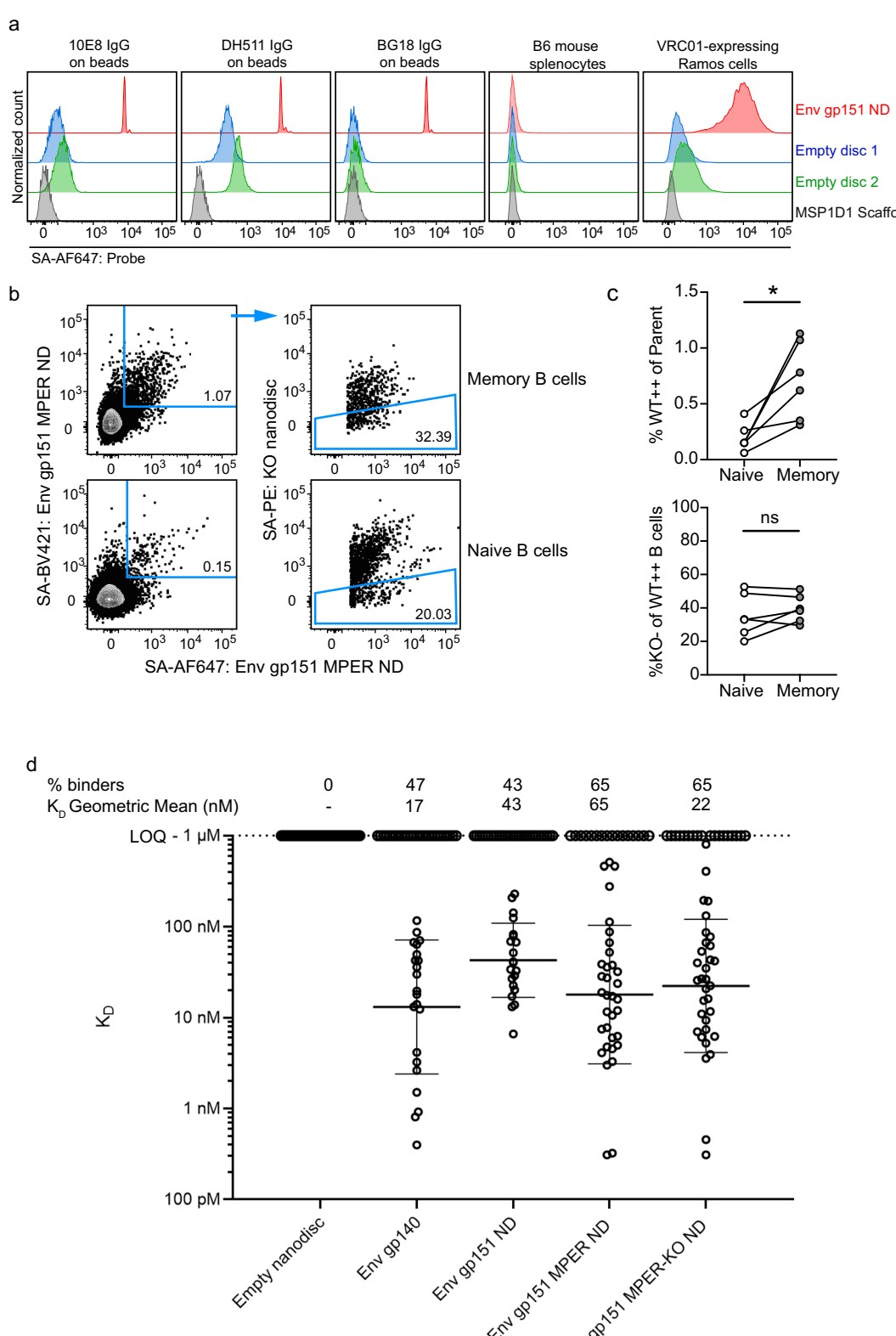

MPER KO version, and overall similar geometric mean affinity to the tested Env nanodiscs provided evidence that the elicited antibodies were not targeting the MPER peptide but bound to the exposed base epitope or other off-target epitopes. Nevertheless, using the nanodiscs as FACS sorting probes and in SPR, we were able to sort and measure affinities of several antibodies that were specific to the transmembrane Env gp151 version (e.g., Ab_38, Ab_48, Ab_79; Supplementary Fig. 3b).

Non-human primates (NHPs) are an important model for late-stage pre-clinical vaccine development and are often used as final validation before clinical trials. *Rhesus macaques*, like humans, have extensive genetic diversity within the immunoglobulin loci and are also capable of making long HCDR3s[47–49]. To test whether nanodisc-based FACS tetramer probes could be successfully used to assess RM B cell responses, experiments were carried out using PBMCs from RMs

**Fig. 4 | HIV Env nanodisc FACS probe validation and pilot use in pre-clinical mouse model. a** Nanodiscs were tested for binding to HIV MPER bnAbs 10E8 and DH511 and gp120-specific bnAb BG18 coupled to FACS compensation beads. B6 mouse splenocytes were used as negative control cells, and VRC01-expressing Ramos cells as positive control cells. Empty nanodiscs were assembled with either DOPC lipids (blue) or a mixture of neutral and charged lipids (green). Recombinant scaffold protein MSP1D1 without nanodisc assembly was included as a negative control (gray). **b** Memory (CD19⁺IgM⁻IgD⁻) and naïve (CD19⁺IgM⁺IgD⁺) B cells from mice 6 weeks after immunization with mRNA-LNPs encoding Env gp151 MPER ND were analyzed by flow cytometry. Antigen-specific (Env gp151 MPER ND⁺⁺) and epitope-specific (KO⁻) B cells were detected using immunogen-matched WT and 10E8-epitope KO nanodisc probes, and WT nanodisc⁺⁺ cells were sorted for BCR sequencing. The complete gating strategy is shown in supplementary Fig. 3. **c** Percent antigen-specific memory B cells in the naïve vs. memory compartments of each immunized mouse (top), and proportion of 10E8 epitope-specific (KO⁻) cells within each antigen-specific population. Symbols represent individual animals. *$p = 0.018$, ns: $p = 0.26$, two-tailed paired *t*-test. **d** Sorted cells were sequenced, and selected antibodies were purified for determining apparent affinities by nanodisc SPR modality B. Symbols indicate individual antibodies, bars geometric mean of binders and error bars geometric standard deviation of binders. Source data are provided as a Source Data file.

immunized previously with soluble Env MD39[50]. PBMCs were stained with the nanodisc probes to determine whether there was substantial background binding to the lipid component of the nanodisc probes. Pre-immunization PBMCs showed low background binding of IgD⁻ memory B cells to nanodisc, while PBMCs from 2-weeks post an MD39 booster immunization showed high binding to the nanodisc probes (Fig. 5a). Next, the nanodisc probes were compared directly to soluble Env gp140 protein probes to characterize B cells from RMs that had been immunized with MD39.3 gp151 mRNA (matching Env gp151 ND sequence) 4-weeks prior[36]. Across two animals, there were no differences in the frequency of tetramer⁺⁺ cells between the Env gp151 ND nanodisc and Env gp140 soluble protein probes (Fig. 5b). Taken together, these data show that nanodiscs bearing stabilized HIV Env constructs can be deployed as tetramer probes for the evaluation of pre-clinical HIV B cell responses in NHPs.

## GP nanodiscs in cryo-EM structural studies

To demonstrate how structural studies of nanodisc GPs can inform vaccine design, we set out to determine the structure of prototype MPER vaccine candidate Env gp151 MPER ND in complex with the 10E8 Fab by cryo-EM. We included two other HIV bnAbs: VRC01, which recognizes the CD4 binding site, and BG18, which targets the N332 glycan supersite. This antibody cocktail complex serves three purposes: First, the added Fabs improve the orientation distribution of particles on cryo-EM grids. Second, they may reduce the chances of freeze-denaturation by preventing particles from adhering to the air-water interface during grid preparation. Third, determining the structure of a membrane-embedded trimer in complex with multiple bnAbs targeting different sites would assist the development of a protective HIV vaccine that will likely require the elicitation of bnAbs targeting several sites to elicit sufficient neutralization breadth and potency to counter global Env diversity. A trimer boost would benefit from being able to engage three or more sites (e.g., N332 supersite/V3-glycan, CD4-binding site, and MPER, to which Apex/V2-glycan and/or fusion peptide could also be included) in a single immunogen. The high-resolution structure of the 10E8 epitope has thus far been resolved only with x-ray crystallography and in complex with MPER peptide alone[51–54]. After confirming features typical of nanodiscs with trimeric GP with or without Fabs in ns-EM 2D class averages, we proceeded to cryo-EM analysis (Supplementary Fig. 4a, b). From our cryo-EM data, we produced two reconstructions: the complex with two 10E8 Fabs at 4.3 Å, and the complex with a single 10E8 Fab at 3.5 Å resolution, which enabled atomic-level interpretation. We noted that while 10E8, VRC01, and BG18 were all added at the same step after the nanodisc batch assembly at ~10x molar excess, all VRC01 and BG18 sites were occupied, but the 10E8 binding site was partially underoccupied in all reconstructions, suggesting steric hindrance of the MPER epitope with increasing 10E8 occupancy (Fig. 6a–c). In all reconstructions, 10E8 is wedged between the Env ectodomain and the nanodisc lipid bilayer. The MPER antibody-induced Env tilt angle was evaluated by aligning the low-pass filtered maps to the nanodisc representing the bilayer plane. A single 10E8 Fab appears to be accommodated with minimal Env tilting while binding of a second

10E8 Fab pushes Env into a more tilted position (Fig. 6b). Unlike in our previous WT Env AMC011 nanodisc cryo-EM reconstruction, density for the TM domain remained undetected. This may have been due to the R696S mutation disrupting the stabilizing connection point of the three helices in the middle of the bilayer[29]. Additionally, truncation of the CT may have allowed the TM helices to adopt less defined positions within the bilayer. The class with a single 10E8 Fab bound was subjected to 3D variability analysis (3DVA), which revealed distinct conformations of the MPER helix with ~30° lateral twist of the Fab in conjunction with an apparent dislocation of the adjacent protomer (Supplementary Fig. 4c, d; Supplementary video 1).

Using the model of Env gp151 MPER ND complex, we were able to resolve novel structural details of the 10E8 epitope, including a binding pocket for 10E8 antibody heavy chain framework region 3 (HFR3), formed by Env gp120 and gp41 subunits (Fig. 6d, inset 5). We identified 14 aa in 10E8 contacting the C terminus of gp120, helices α6 to α8 of gp41 in protomer 1, and L660 of gp41 in protomer 2. These contacts are either in the binding pocket or in the helices forming a collar around the termini of gp120. Strikingly, this binding pocket appeared to be targeted by the HCDR3 of another HIV bnAb, 3BC315, that has been shown to accelerate trimer dissociation[55]. To confirm this similarity in engaging the binding pocket, we determined the cryo-EM structure of Env gp140 (BG505 MD39.3) in complex with BG18, VRC01, and 3BC315 Fabs to 3.1 Å resolution (Fig. 6f; Supplementary Fig. 4f, g). 3BC315 engages the binding pocket mainly with H100A, G100B and Y100E at the tip of the HCDR3. In 10E8, the interaction is driven by I75, S74 and N76 of HFR3 (Fig. 6d, inset 5). The observed similarity of the binding pocket engagement led us to propose that these antibodies may share a common mechanism for trimer destabilization. As part of the binding pocket, both antibodies contact W623 in gp41 (Fig. 6d, inset 5; and Fig. 6f), which is >99% conserved across >5,000 diverse HIV strains from the LANL database (http://www.hiv.lanl.gov), and has previously been shown to be part of a tryptophan clasp contributing to a stabilizing four-helix collar around the termini of gp120[56]. Engaging the binding pocket leads to widening of the gp41 protomer-protomer gap (Fig. 6g), which would eventually lead to destabilization of the contacts in the collar around the termini of gp120 and premature triggering of the fusion peptide. This would normally occur during virus entry-associated Env rearrangements that follow CD4 receptor binding.

We examined the functional importance of the contacts identified from the cryo-EM structure by conducting pseudovirus neutralization assays (Fig. 7a). Reversions of mature 10E8 HFR3 paratope residues to their predicted germline sequences all reduced neutralization activity, with F77T resulting in approximately 8-fold reduction, suggesting that the identified contacts contribute to 10E8 neutralization potency (Fig. 7b). Mutations within the 10E8 epitope were introduced to the BG505 T332N Env sequence commonly used as a control in HIV neutralization panels. Mutating the key MPER residues (N671A, W672A and K683A) engaged by the 10E8 HCDR3 completely abrogated neutralization, as expected[57]. Mutating 10E8 contact residues visualized in the new structure also showed substantial effects on 10E8 neutralization. Mutations R500A, T536A, L619A, S620A, D624A and N625A each

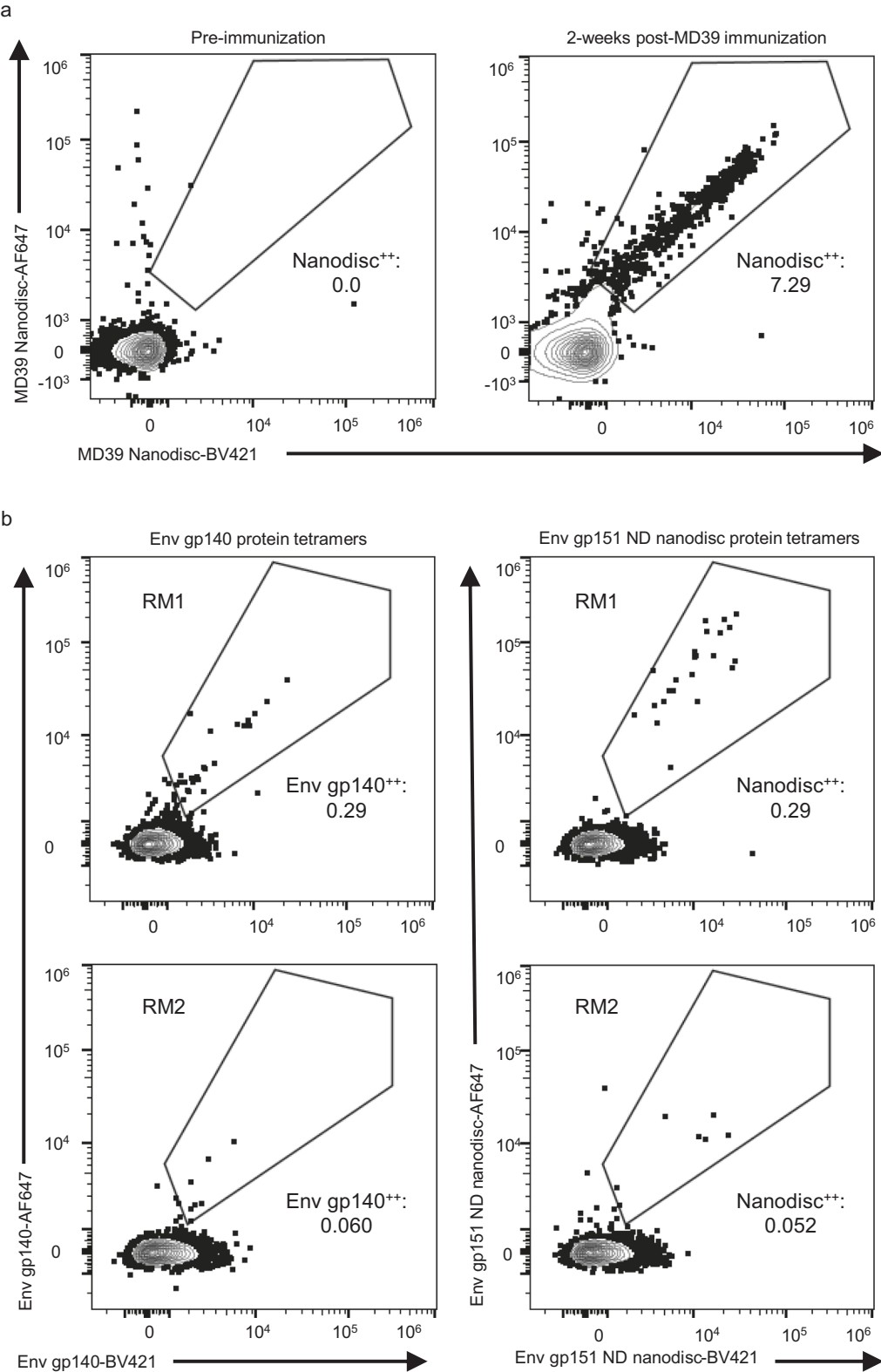

**Fig. 5 | HIV Env nanodisc probe validation for sorting NHP cells. a** Flow cytometry plots showing CD20⁺IgD⁻ memory B cells stained with Env gp151 ND nanodisc on two different fluorophores. Pre-immunization and 2-weeks post-immunization PBMCs from the same RM are shown, and the frequency of Nanodisc⁺⁺ cells of IgD⁻ cells is listed. **b** Flow cytometry plots showing CD20⁺IgG⁺ memory B cells stained with either soluble Env gp140 protein or Env gp151 ND nanodisc on two different fluorophores. PBMCs from two different RMs 4-weeks post mRNA Env gp151 immunization were split and stained with Env gp140 protein tetramers (left column) or matching nanodisc Env gp151 ND tetramers (right column). The frequency of tetramer⁺⁺ cells of IgG⁺ cells is listed for each plot.

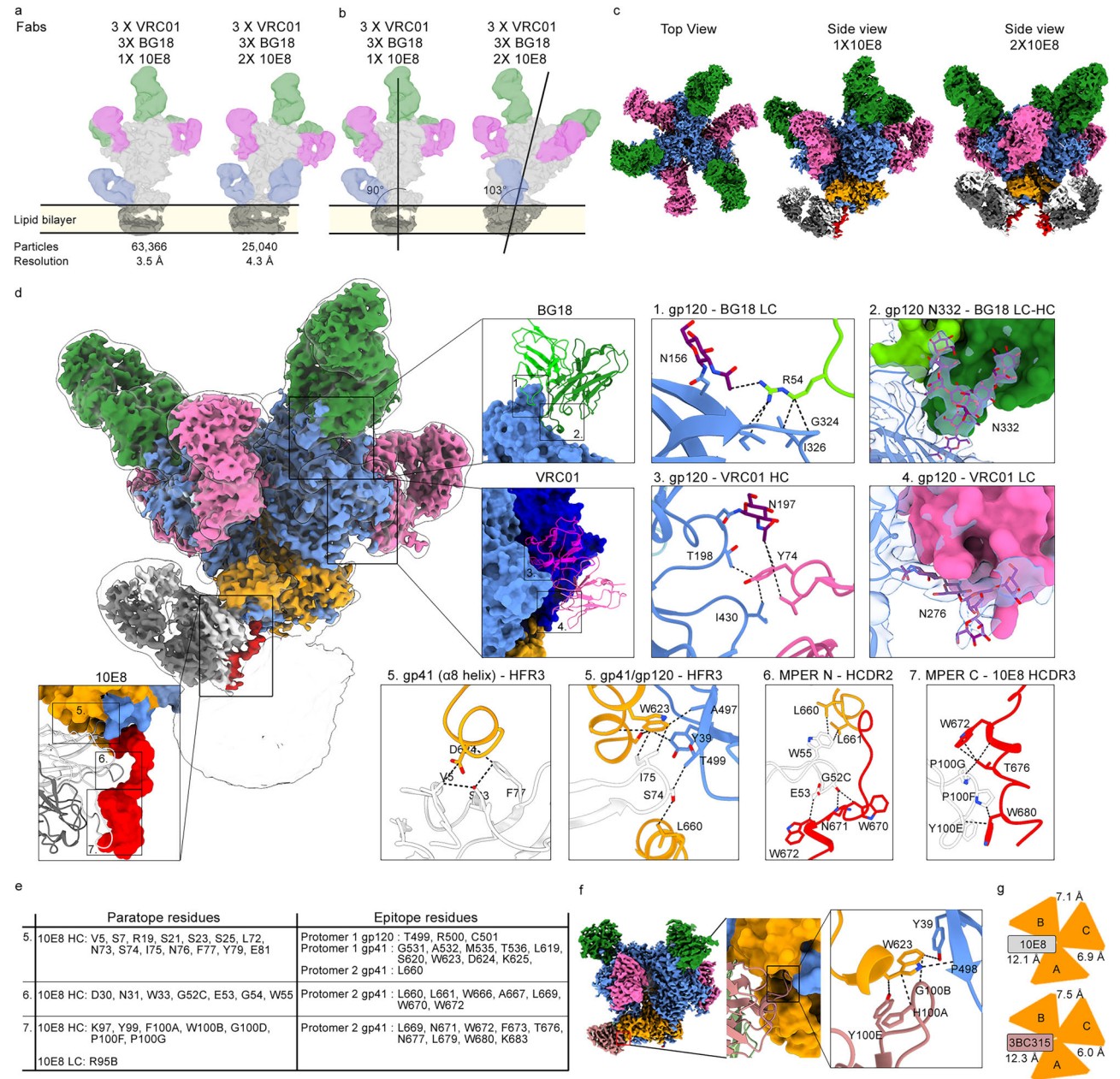

**Fig. 6 | Structure of Env gp151 MPER ND in complex with bnAbs BG18, VRC01 and 10E8 resolved by cryo-EM. a** Low-pass filtered map of different Fab occupancy states and the location of the bilayer. BG18 Fab is highlighted in green, VRC01 in pink and 10E8 in blue. **b** 90° clockwise rotated side views showing the Env tilt angle in relation to the lipid bilayer surface. **c** Top view and side views of the highest resolution maps of both 10E8 occupancy states. Densities are highlighted as follows: gp120 in blue, gp41 in orange, BG18 in green, VRC01 in pink, HC of 10E8 in white, LC of 10E8 in dark gray and MPER peptide in red. **d** The highest resolution reconstruction with a single 10E8 was used for model building. Insets show interfaces of the three bound bnAbs and key amino acid and glycan interactions with BG18 HC (dark green) and LC (light green) in insets 1–2, VRC01 HC (magenta) and LC (pink) in insets 3-4, and 10E8 HC (white) and LC (dark gray) in insets 5-7. Glycans are highlighted in purple. Insets 2 and 4 exemplify the electron density used to build interacting glycans (transparent blue. **e** Epitope-paratope analysis of 10E8 binding interface shown in d. Contacting residues are separated into three components corresponding to insets 5–7. **f** Binding pocket of 3BC315 (salmon) HCDR3 with key interactions highlighted. **g** gp41 protomer distance analysis in the nanodisc complex structure with 10E8 Fab, and soluble Env structure with 3BC315 showing widened interface between protomers A and B due to antibody binding.

led to modest (1.5- to 2.7-fold) decreases in neutralization sensitivity to 10E8 (Fig. 7c). T499G and M545A resulted in modest (2-fold) increases in sensitivity to 10E8. Notably, mutation L660A resulted in a 7.5-fold increase in 10E8 neutralization sensitivity (IC$_{50}$ from 0.15 to 0.02). Thus, the new complex structure identified multiple residues that impact 10E8 neutralization potency.

Another novel structural feature of the 10E8 antibody was the dynamic remodeling of the N-terminal MPER helix, most likely driven by interactions with HCDR1 (D30, N31 and W33) and HCDR2 (G52C,

G53, G54, W55, Fig. 6d, e; Supplementary video 1). The root mean square deviation (RMSD) of this HCDR1 and HCDR2 10E8 paratope region is 1.52 Å compared to a published x-ray structure of 10E8 Fab in complex with a gp41 peptide (PDB 4G6F). In contrast, the RMSD of HFR3 between the nanodisc cryo-EM complex and peptide-bound crystal structure is 0.46 Å. Although the 10E8 crystal structure fit well into the cryo-EM map (Supplementary Fig. 5a), we calculated an RMSD of 9.59 Å of the entire MPER domain between the two structures. The difference between the MPER in the two structures stems

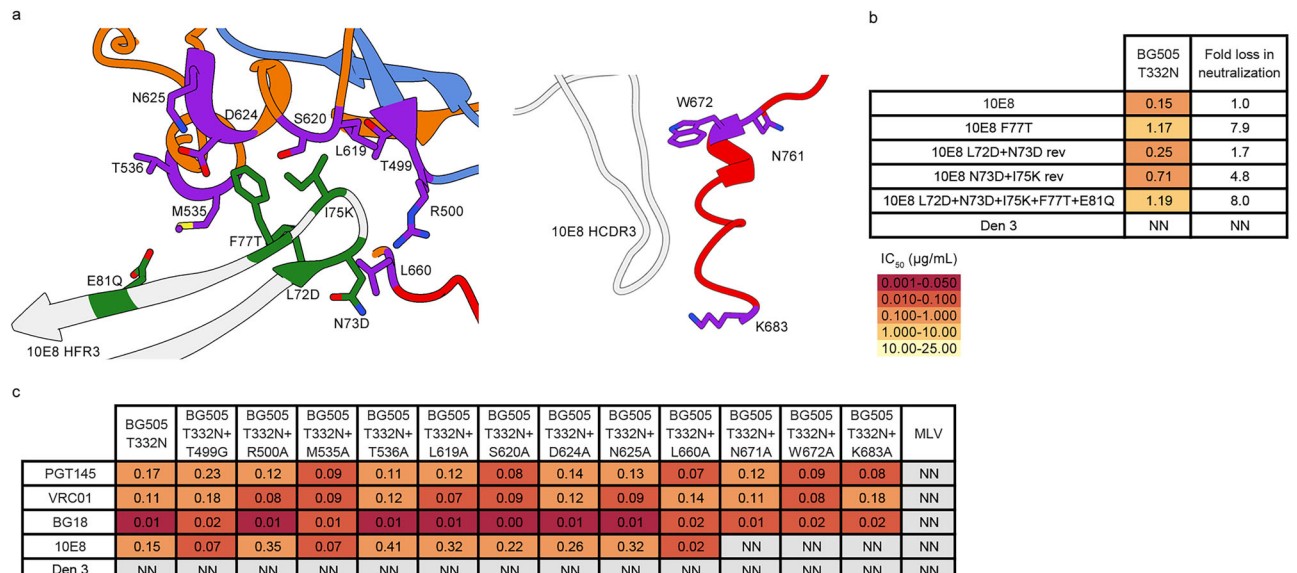

**b**

| | BG505 T332N | Fold loss in neutralization |
|---|---|---|
| 10E8 | 0.15 | 1.0 |
| 10E8 F77T | 1.17 | 7.9 |
| 10E8 L72D+N73D rev | 0.25 | 1.7 |
| 10E8 N73D+I75K rev | 0.71 | 4.8 |
| 10E8 L72D+N73D+I75K+F77T+E81Q | 1.19 | 8.0 |
| Den 3 | NN | NN |

$IC_{50}$ (µg/mL)
0.001–0.050
0.010–0.100
0.100–1.000
1.000–10.00
10.00–25.00

**c**

| | BG505 T332N | BG505 T332N+ T499G | BG505 T332N+ R500A | BG505 T332N+ M535A | BG505 T332N+ T536A | BG505 T332N+ L619A | BG505 T332N+ S620A | BG505 T332N+ D624A | BG505 T332N+ N625A | BG505 T332N+ L660A | BG505 T332N+ N671A | BG505 T332N+ W672A | BG505 T332N+ K683A | MLV |
|---|---|---|---|---|---|---|---|---|---|---|---|---|---|---|
| PGT145 | 0.17 | 0.23 | 0.12 | 0.09 | 0.11 | 0.12 | 0.08 | 0.14 | 0.13 | 0.07 | 0.12 | 0.09 | 0.08 | NN |
| VRC01 | 0.11 | 0.18 | 0.08 | 0.09 | 0.12 | 0.07 | 0.09 | 0.12 | 0.09 | 0.14 | 0.11 | 0.08 | 0.18 | NN |
| BG18 | 0.01 | 0.02 | 0.01 | 0.01 | 0.01 | 0.01 | 0.00 | 0.01 | 0.01 | 0.02 | 0.01 | 0.02 | 0.02 | NN |
| 10E8 | 0.15 | 0.07 | 0.35 | 0.07 | 0.41 | 0.32 | 0.22 | 0.26 | 0.32 | 0.02 | NN | NN | NN | NN |
| Den 3 | NN | NN | NN | NN | NN | NN | NN | NN | NN | NN | NN | NN | NN | NN |

**Fig. 7 | Effect of 10E8 epitope and paratope mutations on HIV pseudovirus neutralization. a** Locations of 10E8 epitope (purple) and paratope (green) residues selected from the cryo-EM structure. **b** 10E8 paratope residues were reverted to the corresponding germline sequences, and the neutralization potencies of the resulting antibodies were compared to mature 10E8 bnAb. Raw $IC_{50}$ values as determined by the TZN-bl assay, and fold changes in $IC_{50}$ compared to 10E8 are indicated. **c** Neutralization potency of 10E8 and control antibodies PGT145, VRC01, and BG18 against pseudoviruses with mutations in the 10E8 epitope. Dengue virus type 3 (Den 3) antibody and Murine Leukemia (MLV) pseudovirus were included as negative controls.

predominantly from the N-terminal portion of the MPER that is connected to the rest of the ectodomain in the nanodisc Env cryo-EM structure. Further comparison of these structures showed that when aligned to the MPER peptide, 10E8 in the crystal structure adopts a ~25° rotated position (Supplementary Fig. 5). Although they engage the same residues within the MPER helix, only 14 of 29 contacting residues in the paratope and 15 of 26 in the epitope were observed in the crystal structure as compared to the cryo-EM structure. Comparison of the cryo-EM complex structure with crystal structures of other MPER bnAbs bound to MPER peptides showed that other MPER bnAbs can accommodate the same approximate space between the lipid bilayer and ectodomain as 10E8 but adopt different binding poses. This suggests that although the MPER peptide contacts remain conserved, variations in interactions are likely outside the core MPER epitope, and MPER bnAbs can adopt a wide range of binding modes, which is also evident in our cryo-EM data (Supplementary video 1). Taken together, the comparison to previous MPER peptide-bnAb crystal structures suggests that the native membrane environment of Env is required for complete structural characterization of MPER targeting bnAbs. Our observations of the native, dynamic binding mode of the 10E8 bnAb and structural information of the entire side chain contact network will inform the development of next-generation MPER targeting vaccines.

## Discussion

Here we present a rigorously optimized glycoprotein nanodisc platform for vaccine design. We show that the assembled GP nanodiscs enable characterization of transmembrane glycoprotein vaccine candidates with key methods in rational, iterative vaccine development that have traditionally been available only to soluble mimics. Applicability of the platform is demonstrated by using SPR to measure antibody kinetics, FACS to characterize new vaccine designs and B cell responses from preclinical immunization experiments, and cryo-EM to provide structural guidance for immunogen design.

In addition to confirming the assembly of EBOV GP in nanodiscs with two Ebola-specific antibodies, we characterized the kinetic landscape of several HIV antibodies binding to HIV Env in nanodiscs.

These confirmed that glycan shield modifications and a transmembrane-domain point mutation identified through directed evolution increased MPER-targeting bnAb 10E8 affinity to prototypical HIV Env MPER-targeting immunogen Env gp151 MPER ND as compared to its design predecessor. The 10E8 affinity was previously reported to be ~2.8 nM when measured by SPR to an MPER peptide embedded in a lipid bilayer. This value is remarkably close to the affinity of 10E8 for Env gp151 MPER ND in nanodisc measured here (3.6 nM), suggesting that the engineering removed the steric barriers provided by glycans[58]. It should be noted, however, that for membrane proximal epitopes, lipid bilayer surface properties, native glycoprotein flexibility, and bivalency of IgGs may all affect the antibody-antigen interaction. GP nanodiscs provide a tool for future studies to examine these effects in detail.

While the in vivo responses to the engineered immunogen in a humanized mouse model appeared to be outside the targeted C-terminal region of MPER, the identified antibodies will provide guidance for further immunogen engineering. Potential targets of the extracted antibodies include the engineered glycan hole or off-target epitopes at the trimer base. The base of the trimer is partly occluded by the membrane, but some off-target epitopes may still be accessible. This was also indicated by EM-based polyclonal epitope mapping (EMPEM) showing that transmembrane gp151 Env administered as mRNA still elicits base-binding antibodies[15,36]. Importantly, using the biotinylated GP nanodisc as a FACS probe, we were able to extract antibodies targeting sites that are not fully present in the soluble form of the immunogen. Although NHP immunization was beyond the scope of this study, we also confirmed that GP nanodiscs can be used to identify antigen-specific responses from NHPs, facilitating future studies. Two promising approaches for overcoming the challenges in eliciting MPER-targeting antibodies have been recently described, namely priming with germline-targeting immunogen 10E8-GT12[24,46] or immunizing with MPER peptide alone in liposome formulation[25]. Both solutions present a more accessible MPER peptide for antibody HCDR3. These immunogens can be followed by a next-generation Env boost immunogen presenting the complete epitope to guide the maturing antibody toward breadth and higher neutralization potency.

The analyses and methods presented here provide vital guidance for the design of such transmembrane booster immunogens.

The platform enabled us to determine the structure of Env gp151 MPER ND to 3.5 Å, leading to the discovery of structural features that will inform the next generation of immunogens. The structure confirms that three HIV bnAb binding sites targeted in ongoing vaccine development projects can be engaged by the same immunogen[24,47,59]. This further demonstrated that the nanodisc platform has the potential to be used in cryo-EMPEM to resolve structures of membrane proximal antibody epitopes from a mixture of specificities from post-immunization serum[60]. As multiple epitopes are now being engineered into Env trimers for germline targeting, it is critical to understand if and how they may compete or synergize. Additional value in using GP nanodiscs is that, in contrast to previous structural studies using soluble mimetics of Env[61,62], this nanodisc-based structure more closely resembles the native transmembrane protein immunogen, which may be important when assessing glycan engagement in the epitope. Glycan analysis of Env gp151 ND and Env gp151 MPER ND agreed with earlier studies showing that when Env is expressed as a transmembrane protein, glycan processing more closely resembled that of the virus particle with a higher proportion of complex-type glycans compared to soluble forms[26,27]. The most significant value of the Env gp151 MPER ND complex cryo-EM structure, however, is in characterizing the MPER epitope in unprecedented detail. We were able to resolve contacts of 10E8 bnAb with the Env ectodomain that have thus far remained elusive in structural studies. A binding pocket for 10E8 HFR3 in the gp120-gp41 interface of the ectodomain, and HCDR1, 2 and 3 contacts to the entire MPER domain ending at the lipid bilayer interface, form a continuous network of side chain interactions together spanning 29 aa in the paratope and 26 aa in the epitope. Similar analysis of a previously determined crystal structure of 10E8 bound to a C-terminal MPER peptide (PDB 4G6F) only revealed part of these interactions, including 14 aa in the paratope and 15 aa in the epitope[54]. These details can now be used as a structural template to design immunogens for improved engagement of MPER antibodies. Specifically, neutralization assays indicated that the L660A mutation may improve 10E8 MPER antibody engagement; a similar effect was previously indicated for HIV MPER targeting antibodies 4E10, 2F5 and Z13e1[63]. Finally, 3D variability analysis of the cryo-EM data illustrates significant remodeling of the N-terminal MPER helix induced by HCDR1 and HCDR2 of 10E8. While care should be taken when interpreting flexible regions, this captures detailed snapshots of the native flexibility of the glycoprotein and MPER domain. The data depict the range of flexibility and may offer clues on how to accommodate native glycoprotein flexibility in vaccine design.

In conclusion, the GP nanodisc platform offers a scalable and reproducible solution that provides a more complete and native-like environment for transmembrane glycoprotein vaccine designs. This platform can therefore accelerate the development of next-generation vaccines and may be applicable beyond vaccine development. The described methods present significant additions to the rapidly increasing nanodisc applications in drug discovery technologies[64]. They may be used in workflows for small molecule screening, antibody discovery, and functional studies involving interactions with both extracellular and intracellular binders of any transmembrane protein.

## Methods

### Ethical statement
This work complies with all relevant ethical regulations. All animal procedures and experiments were performed under the approval of Scripps Research IACUC 20-0001-2.

### Protein expression and nanodisc assembly.
This method description aims to provide a complete platform that enables routine use of the nanodisc approach for iterative vaccine design using viral glycoproteins. While retaining the versatility of a modular approach, each step is simplified and validated for maximizing reproducibility and scalability. All glycoprotein and antibody constructs were codon optimized for expression in human cells and synthesized and cloned into expression vectors pHLSec (glycoproteins) or its variants pCW-CHIg-hG1, pFabCW, pCW-CLig-hL2 or pCW-CLig-hk (IgG heavy chains, Fab heavy chains, lambda light chains and kappa light chains, respectively) by GenScript Biotech. Antibodies were expressed and purified from HEK293F (Thermo Fisher Scientific, #R79007) Ramos cells according to the manufacturer's instructions using rProtein A Sepharose Fast Flow resin (Cytiva, #17127903) as described earlier[24,47]. Antibodies 10E8, EBOV-296, EBOV-13C6, PGZL1, DH511, BG18, VRC01, 4E10, PGZL1 H4K3, PGZL1 gVmDmJ, VRC42.01, VRC43.01, 35022, PGT128, PGT145, 10E8 UCA, 3074, B6, F105, RM19R and RM20A3 were expressed in house.

Soluble Env GP was purified as described earlier[61]. The transmembrane glycoprotein purification method was modified from the earlier protocol[29]. For one liter of transfected cells, 1 mg of DNA was mixed with 25 ml of OPTI-MEM medium (Thermo Fisher Scientific, # 31985062) and sterile filtrated through 0.22 μm filter unit. 3 mg of PEI MAX (Polysciences, #24765) was separately mixed with 25 ml of OPTI-MEM, filtered and then combined with the DNA. Combined DNA and PEI MAX were then added to HEK293F cells at 1 million/ml density, and cells were incubated at 37 °C, 8% CO₂, and 80% humidity under shaking at 180 rpm for 3 days. Cells were harvested by centrifugation at 1500 rcf in 4 °C for 15 min, washed with ~300 ml cold PBS, and centrifuged again at 1500 rcf in 4 °C for 15 min. Pelleted cells were lysed with ~5 ml of lysis buffer per gram of cells (50 mM Tris-HCl (pH 7.4), 300 mM NaCl, 0.5% TX-100). Prior to use, approximately 1 protease inhibitor cocktail tablet per 50 ml of lysis buffer, and 2 ml of BioLock (IBA Lifesciences, # 2-0205-050) per liter of cells was added to the lysis buffer. Cells were lysed for 1 h at 4 °C in an overhead rotating mixer. Lysed cells were centrifuged for 1 h at 25,000 rcf, followed by filtration of the supernatant through a 0.22 μm bottle top filtration unit. This resulted in a cleared lysate that was passed through Strep-tactin XT 4Flow matrix (IBA Lifesciences, #2-5010-025) in a gravity flow column. Approx. 400 μl of drained matrix was used per liter of cells. Matrix was then washed three times with 2 ml of wash buffer 1 (50 mM Tris-HCl (pH 7.4), 300 mM NaCl, 0.1% (w/v) CHAPS, 0.03 mg/mL deoxycholate), three times with 2 ml of wash buffer 2 (50 mM Tris/HCl pH 7.4, 500 mM NaCl, 1 mM EDTA, 0.1% n-dodecyl-ß-D-maltoside (DDM), 0.01% cholesteryl hemisuccinate (CHS), 0.03 mg/ml deoxycholate) and three times 2 ml of wash buffer 3 (50 mM Tris/HCl, pH 7.4, 150 mM NaCl, 0.02% DDM, 0.002% CHS, 0.03 mg/ml deoxycholate). After this, the matrix with detergent-solubilized GP was transferred to a 5 ml test tube for the nanodisc assembly step.

Lipid stocks were prepared prior to disc assembly in 1 mM final concentrations. Premade 10% DDM/ 0.1% CHS stock (Anatrace, #D310-CH210) was used throughout the study. All lipids in this study were prepared from chloroform stocks (Avanti Polar Lipids). Chloroform was first evaporated under a gentle nitrogen stream until a transparent film was formed around the glass vial. Lipid film was then rehydrated for 1 h at room temperature in lipid rehydration buffer (50 mM Tris/HCl, pH 7.4, 150 mM NaCl, 0.1% DDM, 0.01% CHS) in a volume bringing the lipid stock concentration to 5 mM. After thorough mixing by vortexing, this stock was diluted further to a final lipid stock solution at 1 mM concentration and sonicated with a microtip (20-25% intensity, 50% time cycles in 4 °C) until clear. Most lipid stock solutions cleared within 10–20min, but with some lipids, the DDM/CHS amount was increased up to 1%/0.01%. 1 mM lipid stocks were aliquoted and stored up to 6 months at −20 °C. Nanodiscs with 80% DOPC (1,2-dioleoyl-sn-glycero-3-phosphocholine) and 20% CHS were assembled to limit possible lipid headgroup charge effects and establish baseline kinetics for MPER antibodies. Alternatively, a lipid mix roughly following HIV particle lipid composition was used containing 45% DOPC, 7% DOPS

(1,2-dioleoyl-sn-glycero-3-phospho-L-serine), 7% DOPA (1-palmitoyl-2-oleoyl-sn-glycero-3-phosphate), 7% DOPE (1-palmitoyl-2-oleoyl-sn-glycero-3-phosphoethanolamine), 7% DOPG (1,2-dioleoyl-sn-glycero-3-phospho-(1'-rac-glycerol)), 5% sphingomyelin, 2% PI(4,5)P2 (L-α-phosphatidylinositol-4,5-bisphosphate), and 20% CHS. Scaffold protein was either produced in-house using standard *E. coli* expression method or purchased from Sigma-Aldrich (membrane scaffold protein 1D1 #M6574 or biotinylated membrane scaffold protein 1D1 BTN #MSP13).

The engineering of strep-tagged construct and batch assembly instead of nanodisc assembly after GP elution from the affinity matrix was considered a critical point, leading to improved nanodisc assembly as compared to our earlier work. With batch assembly, the entire workflow had better reproducibility, and other conditions, such as lipid and scaffold proportions, had less effect on yield and sample quality[29]. Nanodisc batch assembly was done as follows. Strep-tactin matrix with detergent-solubilized GP bound was drained by gravity flow, after which the gravity flow column was capped, and 400 μl of wash buffer 3 was added to the matrix. 50% matrix slurry was then transferred to a 5 ml test tube. Lipids were added at approx. 350× molar excess and scaffold at 6× molar excess in relation to the glycoprotein. Glycoprotein amount bound to the matrix was estimated with a separate initial experiment where the protein was eluted after the last wash step. Minor variations were made to glycoprotein:scaffold and glycoprotein:lipid ratios without a significant difference in the final yield. Matrix with glycoprotein, lipids and scaffold was incubated for 1 h at 4 °C prior to the addition of SM-2 bio-beads (Bio-Rad, #1523920). Bio-beads were activated according to the manufacturer's instructions in methanol, followed by extensive washing with deionized water. Bio-beads were added to approximately 50% v/v with the nanodisc assembly mixture, i.e., if the total volume of the assembly reaction was 1.5 ml, the final volume after bio-bead addition in the 5 ml test tube was ~3 ml. Detergent was then removed by incubation in a rotating shaker overnight at 4 °C. The following day, an additional ~20% of the total reaction volume of fresh bio-beads was added, followed by an additional 1 h incubation at room temperature to ensure complete removal of detergent. Next, the contents of the test tube were transferred to a new gravity flow column with a pipette tip with the first ~2 mm cut off to allow pipetting biobeads and matrix. Mixture was washed in column twice with 4 ml of TBS and eluted with Strep-tactin elution buffer BXT in 500 μl fractions. A280 of eluted fractions were monitored, and fractions containing protein were pooled and concentrated to <500 μl for SEC performed using a Superose 6 Increase 10/300 column (Cytiva) at 0.75 ml/min in TBS. SDS-PAGE and ns-EM analysis were used to confirm the presence of glycoprotein and scaffold protein, a typical disc appearance in ns-EM 2D class averages and glycoprotein on the disc

For structural biology purposes, Fabs were added to the GP nanodiscs after the assembly, while the nanodisc was still bound to strep-tactin matrix. After the second addition of bio-beads, each Fab was added at ~10× molar excess, and the mixture was incubated for an additional 2 h at room temperature. Complex elution and the following steps were identical to those described above for unliganded glycoprotein nanodiscs. Fractions were pooled after SEC, and the sample was concentrated either to 1 mg/ml for FACS, SPR and other purposes, or up to 11 mg/ml for cryo-EM sample preparation.

### Identification of R696S mutation on HIV Env by mammalian directed evolution
A 250 amino acid long site-saturation mutagenesis (NNK) scan of gp41 was synthesized at SGI-DNA. A fragment of the library containing NNKs from positions T606 to F699 (94 positions) was cloned into BG505 MD39.2 gp151. This library was stably integrated into 293T-rtTA3G cell line[65] using the lentivirus-based mammalian display protocol described previously[34,35,61]. After inducing library expression overnight, the cells were stained with 10E8 IgG and PGT145 Fab and those with the highest binding to 10E8 that were also positive for PGT145 were sorted. After three rounds of sorting, the library DNA was extracted and sequenced by Sanger sequencing. 8 of 19 sequences had a mutation at position 696, which was mutated from R to S, A, V, I and L. The most frequent mutation, R696S, was selected for further characterization.

### Mass spectrometry
**Proteinase K treatment and deglycosylation.** HIV Env glycoprotein (when membrane-bound, denatured in 6 M urea) was exchanged to water using Microcon Ultracel PL-10 centrifugal filter. Glycoprotein was reduced with 5 mM tris(2-carboxyethyl)phosphine hydrochloride (TCEP-HCl) and alkylated with 10 mM 2-Chloroacetamide in 100 mM ammonium acetate for 20 min at room temperature (RT, 24 °C). Initial protein-level deglycosylation was performed using 250 U of Endo H for 5 μg trimer, for 1 h at 37 °C. Glycoprotein was digested with 1:25 Proteinase K (PK) for 30 min at 37 °C. PK was denatured by incubating at 90 °C for 15 min, then cooled to RT. Peptides were deglycosylated again with 250 U Endo H for 1 h at 37 °C, then frozen at −80 °C and lyophilized. 100 U PNGase F was lyophilized, resuspended in 20 μl 100 mM ammonium bicarbonate prepared in $H_2^{18}O$, and added to the lyophilized peptides. Reactions were then incubated for 1 h at 37 °C, subsequently analyzed by LC-MS/MS.

**LC-MS/MS.** Samples were analyzed on a Q Exactive HF-X mass spectrometer. Samples were injected directly onto a 25 cm, 100 μm ID column packed with BEH 1.7 μm C18 resin. Samples were separated at a flow rate of 300 nL/min on an EASY-nLC 1200 UHPLC. Buffers A and B were 0.1% formic acid in 5% and 80% acetonitrile, respectively. The following gradient was used: 1–25% B over 160 min, an increase to 40% B over 40 min, an increase to 90% B over another 10 min and 30 min at 90% B for a total run time of 240 min. The column was re-equilibrated with solution A prior to the injection of the sample. Peptides were eluted from the tip of the column and nanosprayed directly into the mass spectrometer by application of 2.8 kV at the back of the column. The mass spectrometer was operated in a data-dependent mode. Full MS1 scans were collected in the Orbitrap at 120,000 resolution. The ten most abundant ions per scan were selected for HCD MS/MS at 25 NCE. Dynamic exclusion was enabled with an exclusion duration of 10 s, and singly charged ions were excluded.

**Data processing.** Protein and peptide identification were done with the Integrated Proteomics Pipeline (IP2). Tandem mass spectra were extracted from raw files using RawConverter[66] and searched with ProLuCID[67] against a database comprising UniProt reviewed (Swiss-Prot) proteome for Homo sapiens (UP000005640), UniProt amino acid sequences for Endo H (P04067), PNGase F (Q9XBM8), and Proteinase K (P06873), amino acid sequences for the examined proteins, and a list of general protein contaminants. The search space included no cleavage-specificity. Carbamidomethylation (+ 57.02146 C) was considered a static modification. Deamidation in the presence of $H_2^{18}O$ (+ 2.988261 N), GlcNAc (+ 203.079373 N), oxidation (+ 15.994915 M) and N-terminal pyroglutamate formation (−17.026549 Q) were considered differential modifications. Data was searched with 50 ppm precursor ion tolerance and 50 ppm fragment ion tolerance. Identified proteins were filtered using DTASelect2[68] and utilizing a target-decoy database search strategy to limit the false discovery rate to 1%, at the spectrum level[69]. A minimum of 1 peptide per protein and no tryptic end per peptide were required, and precursor delta mass cut-off was fixed at 15 ppm. Statistical models for peptide mass modification (modstat) were applied. Census2[70] label-free analysis was performed based on the precursor peak area, with a 15 ppm precursor mass tolerance and 0.1 min retention time tolerance. "Match between runs" was used to find missing peptides between runs. Data analysis using GlycoMSQuant[42] was implemented to automate the analysis. GlycoMSQuant summed precursor peak areas across replicates, discarded

peptides without NGS, and discarded misidentified peptides when N-glycan remnant-mass modifications were localized to non-NGS asparagines and corrected/fixed N-glycan mislocalization where appropriate.

## SPR

The kinetics and binding affinities of antibody-antigen interactions were analyzed using either the ProteOn XPR36 system (Bio-Rad) or Carterra LSA. TBS pH 7.4 (20 mM Tris, 150 mM NaCl) supplemented with BSA at 1 mg/ml without detergents was used as running buffer in all experiments. In ProteOn XPR36, HC30M XanTec sensor chips were utilized. In modalities B and C, anti-human IgG (Fc) antibody (GE, #BR-1008-39) was used for capturing IgG (ligand) at low densities with nanodisc GP as the analyte. In modality A, anti-Strep-tag antibody (pAb, rabbit, GenScript, #A00875) was used for capturing Strep-tagged nanodiscs GP (ligand) followed by GP specific Fab as the analyte. Approximately 6000-8000 response units (RU) of the capture antibody were covalently attached to the sensor surface using EDC-NHS chemistry. For IgG-antigen interaction studies, approximately 50 to 100 RUs of IgGs at a concentration of 0.3 µg/ml were immobilized on each flow cell. For Fab-antigen interaction studies, approximately 300 to 400 RUs of antigen at 10 µg/ml were captured on each flow cell. Analytes were introduced to the flow cell at a rate of 30 µl/min for 3 min, followed by a dissociation phase of 5 min. Regeneration was performed with 1.7% or 0.85% phosphoric acid, each with a contact time of 60–180 s, repeated four times per cycle. Data analysis was conducted using ProteOn Manager software (Bio-Rad), including raw sensorgram processing, interspot referencing, and column double referencing. Equilibrium or kinetic fits were performed using the Langmuir model as needed.

Kinetics and affinity of antibody-antigen interactions on Carterra LSA using CMDP Sensor Chip (Carterra) for capture IgG – flow nanodisc (modality B) were done as follows. Chip surfaces were prepared according to the Carterra software instructions. In a typical experiment, approximately 500–700 RU of capture antibody (SouthernBioTech, # 2047-01) in 10 mM Sodium Acetate, pH 4.5, was amine coupled on the CMDP chip, taking special care with the concentration range of the amine coupling reagents. We used N-Hydroxysuccinimide (NHS) and 1-Ethyl-3-(3-dimethylaminopropyl) carbodiimide hydrochloride (EDC) from the Amine Coupling Kit (GE, #BR-1000-50). Highest coupling levels of capture antibody were achieved by using 10 times diluted NHS and EDC during surface preparation runs as compared to the kit instructions (10ml of water each to give 11.5 mg/ml and 75 mg/ml, respectively, according to the kit instructions). Thus, the concentrations of NHS and EDC were 1.15 mg/ml and 7.5 mg/ml, and the activation time was reduced to 1 min. The concentrated stocks of NHS and EDC were stored frozen in −20 °C for up to 2 months without noticeable loss of activity. The capture antibody was used at a concentration of 25 µg/ml with 10 min contact time. Phosphoric Acid 1.7% was used as a regeneration solution with 60 s contact time and injected three times per cycle. The concentration of ligands was approximately 1 µg/ml, and the contact time was 5 min. Raw sensorgrams were analyzed using Kinetics software (Carterra) with interspot and blank double referencing, and the Langmuir model. Analyte concentrations were quantified on a NanoDrop 2000c spectrophotometer using the absorption signal at 280 nm. Analytes were buffer exchanged into the running buffer using dialysis. In a typical run, we covered a broad range of affinities and used two referencing practices depending on the off-rate of the ligand. For fast off-rate (faster than $9 \times 10^{-3}$ 1/s), we use automated batch referencing that includes overlay y-aline and higher analyte concentrations. For slow off-rates ($9 \times 10^{-3}$ 1/s or less), we use a manual process referencing that includes serial y-align and lower analyte concentrations. After automated data analysis by Kinetics software, we performed additional filtering to remove datasets with the highest response signals smaller than signals from negative controls using an R script.

## FACS

**Validation of Env-Nanodisc proteins as fluorescent baits.** Streptavidin (SA) conjugated-antigen baits were prepared by combining biotinylated Env-nanodisc or empty nanodisc with fluorescent SA at room temperature for at least 1 h in the dark at a bait:SA molar ratio of 2:1. Control beads for FACS were generated by conjugating various bnAbs to compensation beads. Mouse anti-human IgG (BD Biosciences, # 555784) was first captured onto anti-mouse IgK compensation beads (BD Biosciences, # 552843), followed by a wash step and a secondary capture of bnAbs of interest. VRC01-expressing Ramos cells[45], C57BL/6 splenocytes (The Jackson Laboratory, RRID:IMSR_JAX:000664), and the prepared bnAb-conjugated beads were incubated with the nanodisc GP:SA baits at a bait concentration of 10-50 nM in the final staining volume for 30 min at 4 °C in the dark. Data were acquired on a BD FACS Melody and analyzed in FlowJo v10 (FlowJo, LLC).

**Mouse immunization studies.** hD3-3/JH6 mice were immunized as previously described at 2–4 months of age[24]. Mice were maintained on a 12-h white light cycle, with lights on from 06:00 AM to 07:00 PM. The room temperature was kept between 20 °C and 26 °C, and the humidity was maintained between 30% and 70%. All six mice were male based on availability. Briefly, homozygous hD3-3/JH6 mice were injected with 10 µg (50 µl total volume) of Moderna mRNA LNPs encoding Env gp151 MPER I.M. under anesthesia (5% isoflurane induction) in the left quadriceps muscle. After six weeks, mice were euthanized with compressed $CO_2$ (100%) in a clear chamber to allow for visualization of respiration and subsequent death via respiratory cessation. Blood was collected from the chest cavity prior to the removal of the spleen and lymph nodes (mesenteric, inguinal, and popliteal). Tissues were placed in 3 ml FACS buffer (1× PBS Ca/Mg$^{++}$ free, 1 mM EDTA, 25 mM HEPES, pH 7.0, 1% heat-inactivated FBS) on ice, and tissues were disassociated using the rough ends of two sandblasted microscope slides, followed by centrifugation (460 rcf for 5 min at 4 °C). Red blood cell lysis was performed using 1 ml of ACK buffer (Quality Biological) for 2 min on ice before lysis was halted by adding 14 ml FACS buffer per sample. Post lysis and centrifugation (460 × g for 5 min), cells were resuspended in 3 ml Bambanker freezing medium (Bulldog Bio) prior to filtration through a cotton-plugged, borosilicate pasteur pipette into a borosilicate glass test tube. 1 ml filtered-cell solution was subsequently divided into three cryovials/mouse, which were precooled in a styrofoam rack on dry ice. Cells were stored at −80 °C for 2–7 days prior to long-term storage in liquid nitrogen. All work followed IACUC guidelines associated with animal protocol number 20-0001-2.

**Mouse sample preparation and B cell sorting.** Mouse frozen splenocytes and lymphocytes were thawed and stained as previously described[24,59]. Briefly, after thawing and counting, total B cells were isolated by negative selection using the EasySep Mouse Pan-B Cell Isolation Kit (StemCell, #19844) according to the manufacturer's instructions. SA-conjugated baits were prepared as described above. Wildtype antigen baits were conjugated to BV421-SA (BioLegend, #405225) and AlexaFluor 647-SA (Invitrogen, #S21374), while epitope knockout (KO) baits were conjugated to hashtagged TotalSeq-C PE SA (BioLegend, #405261). Purified B cells were stained at 4 °C in dark, with appropriate baits and antibody master-mix consisting of FITC anti-CD19 (BioLegend, #152404), BV786 anti-IgM (BD Biosciences, #743328), PerCP-Cy5.5 anti-IgD (BD Biosciences, #564273), APC-Cy7 anti-F4/80 (BioLegend, #123118), APC-Cy7 anti-CD11c (BD Biosciences, #561241), APC-Cy7 anti-Ly6C (BD Biosciences, #557661), APC-H7 anti-CD8a (BD Biosciences, #560182), and APC-H7 anti-CD4 (BD Biosciences, #560181), all used at 1:100 dilution and prepared in FACS buffer (1% v/v heat inactivated FBS, 1 mM EDTA, 1 mM HEPES in 1× DPBS). The KO bait was first added to cells with the antibody master mix for 15 min, followed by the addition of WT baits for an additional

30 min. Each bait was used at a final bait concentration of 100 nM in the staining volume or 0.5 μg per sample. During the addition of the antibody master mix, a unique TotalSeq-C anti-mouse hashtag antibody (BioLegend) was added to each sample at a concentration of 2.5 μl/up to 20 million cells. Following antibody staining, 1 ml of 1:300 live/dead stain diluted in FACS buffer (Invitrogen, #L34966) was added to each sample for an additional 15 min at 4 °C, then washed with 10 ml of FACS buffer. Cells were resuspended in a final volume of 500 μl FACS buffer and sorted on a BD FACS Melody. Cells were bulk sorted into a 4 °C chilled PCR plate well containing 20 μl of 0.2 μm filtered FBS, and processed for BCR sequencing via the 10× Genomics Single Cell Immune Profiling according to kit protocols, with minor modifications outlined previously[71]. Pooled libraries were sequenced on an Illumina NextSeq 2000 using a 100-cycle P3 reagent kit (Illumina, #20040559).

**BCR sequence analysis.** Sequence analysis Raw sequencing data were demultiplexed, processed into assembled VDJ contigs and counts matrix files, and assigned to specific animal IDs based on TotalSeq-C antibody hashtag counts using Cell Ranger (v6.1) and scab as previously described[71]. Gene assignment, annotation, and formatting into Adaptive Immune Receptor Repertoire (AIRR) format[72] for paired heavy and light chain antibody sequences were performed using Sequencing Analysis and Data library for Immunoinformatics Exploration (SADIE) with a previously described custom germline reference database that included all known mouse antibody genes plus the human IGHD3-3 and IGHJ6 genes that were knocked into hD3-3/JH6 mice[24]. Eight antibodies from each animal were randomly selected for in vitro analysis, except for one animal from which only two total heavy/light pairs were recovered. In addition, 12 antibodies with long HCDR3s (>19 aa) were selected for in vitro characterization.

**NHP flow cytometry.** Frozen NHP PBMC samples were thawed in RPMI media with 10% FBS, 1% GlutaMAX, and 1% Penicillin/Streptomycin (R10). NHP samples used in this study were from animals that were reported in previous studies[36,50]. The recovered cells were then stained with the appropriate antibody panel. Fluorescent antigen probes were constructed by combining fluorophore-conjugated streptavidin and biotinylated nanodisc GP or soluble Env probes with an appropriate volume of PBS in small increments over 45 min at room temperature (RT). Thawed cells were incubated with probes for 30 min at 4 °C. A master mix of surface staining antibodies was then added and incubated for 30 min at 4 °C. Cells were washed and prepared for acquisition on Cytek Aurora (Cytek Biosciences). The following reagents were used in the PBMC staining panel: Alexa Fluor 647 Streptavidin (BioLegend #405237), BV421 Streptavidin (BioLegend #405225), PE Streptavidin (BioLegend #405245), PE-Cy7 Streptavidin (BioLegend #405206), BV711 Streptavidin (BioLegend #405241), BUV615 Streptavidin (BioLegend #613013), Viability eFluor780 (1:2000, Invitrogen #65-0865-14), CD3 BV510 (1:100, BD Biosciences #569488), CD14 BV510 (1:100, BioLegend #301842), CD8a BV510 (1:100, BD Biosciences #563256), CD16 BV510 (1:100, BioLegend #302048), CD20 (1:100, BUV395 BD Biosciences #563782), IgG BV605 (1:100, BD Biosciences #563246), and IgD AF488 (1:50, Southern Biotech #2030-30).

**EM.** Negative stain EM was used for quality control of nanodisc assembly and for initial assessment of antibody complexes for cryo-EM. Glycoprotein nanodisc or nanodisc with threefold molar excess of Fab was applied to a 400 mesh size Cu grid at 0.04-0.06 mg/ml concentration, blotted off with filter paper and stained with 2% uranyl formate for 60 s. NsEM data were collected on a Tecnai Spirit microscope operating at 120 keV, using a Tietz 4k × 4k TemCam-F416 CMOS camera and Leginon automated image collection software[73]. Data was processed using the Relion 3.0 image processing pipeline[66].

For cryo-EM, nanodisc with lipid mix was complexed with 10× molar excess of VRC01, BG18 and 10E8 Fabs after batch nanodisc assembly, while bound to the affinity matrix. The eluted complex was then SEC purified (Superose 6 Increase column, Cytiva), concentrated to 8 mg/ml and frozen on graphene oxide grids (GO on Quantifoils R1.2/1.3, Cu, 400 mesh, Electron Microscopy Sciences) with a 30 s wait time, blot force of 0, and blot time of 2.5-3 s. Fluorinated Fos-Choline-8 (Anatrace # F300F) was added to the sample immediately prior to freezing to a final concentration of 3 mM. GO grids, high sample concentration and fluorinated Fos-Choline-8 were necessary to prevent freeze-denaturation, ensure high particle count per micrograph and prevent orientation bias. 11,871 micrographs were collected at pixel size 0.718 Å using a Thermo Fisher Scientific Glacios 2 microscope operating at 200 kV and equipped with a Thermo Fisher Scientific Falcon 4i camera using a total dose of 44.89 e-/Å2. Automated data collection was performed using the EPU software (Thermo Fisher Scientific), and images were written in the EER frame format. Micrographs were preprocessed using cryoSPARC Live[74], including motion and CTF correction. Briefly, particles were picked with Topaz[75] after training the neural network with a subset of particles first picked with blob picker. Picked particles were subjected to 3 rounds of 2D classification to obtain a final particle stack of 317,792 particles. These were then extracted from micrographs using a box size of 540 pix. Maps containing one or two copies of 10E8 Fab were separated using heterogeneous reconstruction and 3D classification tasks. A class with 7,964 particles containing two copies of 10E8 was refined to 4.3 Å resolution. A class with 111,002 particles containing a single 10E8 was examined with 3D variability analysis[76]. Further classification resulted in a final, cleaned particle stack containing 63,366 particles, which was then subjected to Non-Uniform Refinement[77], leading to a 3.5 Å resolution map, which was then used for model building. We attribute the higher resolution in comparison to our earlier approaches[29] to improvements in nanodisc incorporation efficiency resulting from batch assembly, which in turn allowed the use of Fluorinated Fos-Choline-8 in combination with high protein concentration and graphene oxide grids. These led to an abundance of particles with side-view orientation that are required for high-resolution structure determination. Additionally, improvements in microscope and camera hardware, as well as data processing software, enabled faster grid condition screening and larger datasets. Data collection and processing statistics are summarized in Supplementary Table 2. Model building was initiated by docking in available structures of 10E8 Fab (PDB 4G6F), BG18 Fab (PDB 6DFG), VRC01 Fab (PDB 3NGB) and an AlphaFold3[78]-generated model of the trimer into the map using ChimeraX[79]. Manual adjustment was done using Coot[80], and further refinement was done using Phenix Real Space Refine[81] and Rosetta Relax[82]. Final model statistics are summarized in Supplementary Table 2.

The soluble gp140 (MD39.3) cryo-EM structure was done as follows: VRC01, BG18 and 3BC315 Fabs were added at 4.3× molar excess to purified Env glycoprotein. The complex was purified using a HiLoad Superdex 200 16/600 pg column (Cytiva) and concentrated to 10 mg/ml. Samples were vitrified using a Thermo Fisher Scientific Vitrobot Mark IV operating at 100% humidity, 4 °C and 3–6 s blot times, on UltrAuFoil 1.2/1.3-300 holey gold film grids (Electron Microscopy Sciences). Prior to grid application, lauryl maltose neopentyl glycol (LMNG) was added to a final concentration of 0.005 mM. Imaging was performed using the same microscope and conditions as for Env gp151 MPER ND complex. After initial cryoSPARC particle picking using Blob Picker, templates were selected from 2D classification and used for Template Picker. A total of 886,907 particles were extracted using a box size of 576, downsampled to 144. After additional rounds of 2D classification, an Ab Initio Reconstruction was performed, followed by Non-Uniform refinement. A single 3BC315 Fab was visible in the C1 reconstruction. A mask was created over 3BC315, and a 3D

classification was performed to remove trimers with no 3BC315 particles. A final stack of 243,503 was re-extracted and downsampled to a box size of 400 (resulting in a pixel size of 1.034Å) and subjected to Non-Uniform Refinement, which resulted in a 3.1Å reconstruction. Model building was performed as above, and final model statistics are summarized in Supplementary Table 2.

Maps and models have been deposited to the EMDB and PDB, respectively, under the accession codes listed in Supplementary Table 2.

**TZM-bl Pseudovirus (PSV) Neutralization assay.** PSV were produced in human embryonic kidney 293 T cells (RRID:CVCL 0063) co-transfected using FuGENE 6 (Promega t#E2691) with pseudovirus Env-expressing plasmid and Env-deficient backbone plasmid (PSG3DEnv). PSVs were harvested 72 h post-transfection and kept frozen at −80 °C until use. On the experimental day, viruses were prepared by diluting with D10 media (Dulbecco's Modified Eagle Medium (Gibco #10313-021), 10% FBS (Omega Scientific #FB-02), 1× PenStrep (Gibco #15070-063), and 1× GlutaMAX (Gibco #35050-061) according to previously defined titers. DEAE-Dextran (Spectrum Chemical #DE-132) (5ug/ml final concentration) was added to all PSV except to MLV control due to its potency. Equal volumes of serially diluted mAbs and controls at a starting concentration of 25 μg/ml (Test mAbs and Den3 negative control), 10 μg/ml (VRC01) and 5 μg/ml (BG18) were incubated with HIV PSV in a round-bottom 96-well plate (Costar #3788) at 37 °C for 1 h. TZM-bl cells (RRID:CVCL_B478) were seeded 24 h prior at 100,000 cells/ml in 50 μl/well of half-area 96-well plates (Costar #3688). The plates were then removed from the 37 °C incubator, culture media carefully aspirated, and 25 μl of diluted mAb + PSV added to each well before further incubation at 37 °C for 24 h in a humidified atmosphere of 5% CO2. After 24 h incubation, 75 μl of culture media was added, and the plates were incubated in the same conditions for an additional 48 h. After 48 h (total 72 h), culture media were removed, and cells were lysed with 45 μl/well 1× Luciferase Culture Lysis buffer (Promega #E1531) for 20 min at RT. Neutralization was measured by adding 30 μl luciferase reagent (substrate) per well (Promega #E1501) and measuring luminescence. $IC_{50}$ was calculated using a one-site Fit Log $IC_{50}$ nonlinear regression curve fit constrained from 0-100% in GraphPad Prism 10.6.1.

**Statistics and reproducibility.** Details of statistical analyses can be found in the Figure legends. Statistical analysis using the indicated tests and plotting of all data were performed using Graphpad Prism 10.6.1. Data collection and analysis were not performed blind to the conditions of the experiments. For mouse experiments, the sample size of $n = 6$ per independent experiment was determined by animal availability.

**Reporting summary**
Further information on research design is available in the Nature Portfolio Reporting Summary linked to this article.

## Data availability
All data are available within the main text, the Supplementary Information, or in the following public databases. Cryo-EM coordinates and structures are available in the RCSB Protein Data Bank under PDB IDs 9OGM and 9OGL, and in the Electron Microscopy Data Bank under accession codes EMD-70471, EMD-70470 and EMD-70469. Previously published structures are available under PDB IDs 4G6F (10E8 Fab), 6DFG (MD 39 SOSIP in complex with BG18 Fab) and 3NGB (VRC01 in complex with HIV-1 gp120). Raw and processed BCR sequence reads from immunized mice are available in the European Nucleotide Archive under accession codes PRJEB106503 and OZ390517-OZ391770, respectively, and in the Zenodo public data archive (https://zenodo.org/records/15610801). Mass spectrometry raw data are available in the MassIVE repository under ID MSV000098124. Source data are provided as a Source Data file. Source data are provided with this paper.

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

## Acknowledgements

We thank Nicole Doria-Rose for providing the VRC01-expressing Ramos cell line. We thank Brian Briney for providing access to the Illumina NextSeq 2000 sequencer. We thank Daniel Murin for providing Ebola GP-specific antibodies. We thank Carolyne Kifude for advice on neutralization assay design. This work was supported by National Institute of Allergy and Infectious Diseases UM1 AI144462 (Scripps Consortium for HIV/AIDS Vaccine Development; to S.C., J.C.P., T.S., A.B.W., and W.R.S), R01 AI147826 (to W.R.S. and T.S.), R56 AI192143 (to K.R.); 5F31AI179426-02 (to O.M.S.); Bill and Melinda Gates Foundation Collaboration for AIDS Vaccine Discovery awards (INV-007522 and INV-008813 for the IAVI NAC Center to A.B.W. and W.R.S.; and INV-002916 to A.B.W.); the IAVI Neutralizing Antibody Center (to A.B.W. and W.R.S.); and the Alexander von Humboldt Foundation (to T.S.).

## Author contributions

K.R. designed the nanodisc assembly workflow, performed cryo-EM imaging, data processing and structural analysis, and wrote the manuscript. A.L. and O.K. established conditions and conducted SPR assays. G.O. supervised cryo-EM imaging and data processing and built 3D models. C.F. performed, and J.H.L. and D.S. supervised, establishing conditions for mouse B cell FACS experiments. C.F. performed neutralization assays. J.M.S., O.M.S. and T.S. contributed to immunogen design. S.B. and J.K.D. performed, and J.R.Y. and J.C.P. supervised mass spectrometry glycan profiling. P.J.M. and M.S. performed, and S.C. supervised NHP B cell FACS analysis. S.P. and A.G. contributed, and G.O. and A.B.W. supervised cryo-EM imaging, data processing and model building. D.L. performed nanodisc assembly reactions. P.K. and S.T. performed mouse immunizations. D.L., E.G., R.T., S.E., N.A., D.G. and M.K. produced purified proteins. W-H.L. contributed to cryo-EM sample preparation. S.H. provided mRNA immunogens. T.S., A.B.W. and W.R.S. supervised the study. G.O., T.S., A.B.W. and W.R.S. edited the manuscript. All authors reviewed the manuscript.

## Competing interests

J.M.S. and W.R.S. are inventors on a patent for the BG505 MD39 and N332-GT5 immunogens (US11203617B2 and US20230190914A1). S.H. and W.R.S. are employees and shareholders of Moderna, Inc. All other authors declare no competing interests.
