## [Transparent Peer Review file · Nature Communications]

Virus glycoprotein nanodisc platform for vaccine analytics

Corresponding Author: Professor William Schief

Version 0:

Reviewer comments:

Reviewer #1

(Remarks to the Author)

The manuscript by Schief and colleagues describes the use of nanodiscs to generate soluble membrane anchored viral glycoproteins such as HIV-1 Env as tools for biophysical characterization of vaccine antigens that have been formulated as mRNA LNPs to produce membrane anchored immunogens. Furthermore, nanodisc-Env can be used to analyze antibody responses after vaccination. The authors employ the system to further analyze Env complex formation with the MPER-targeting bnAb 10E8.

Major findings are:

Env gp151 MPER ND with three gp41 glycosylation sites removed and a R696S mutation binds to 10E8 with nanomolar affinity. Removing the glycans helps to increase the affinity from 200 nM to 3.6 nM.

Glycosylation of Env gp151 ND is close to native Env glycosylation.

Biotinylated nanodisc-Env was used as a bait to sort B cells from mice immunized with Env gp151 MPER ND mRNA-LNPs. No MPER-specific responses were detected. Abs were cloned and confirmed no binding to MPER.

Nanodisc-Env can be used as tetramer probes to evaluate B cell responses in NHPs.

CryoEM structures of nanodisc-Env in complex with 10E8 Fabs showed 3:1 and 3:2 complexes, but no 3:3 complexes.

Novel features of the 10E8 epitope include 10E8 contacts with the C-terminus of gp120 and with gp41 helices 6 to 8.

In summary, the described workflow to characterize membrane-anchored viral glycoproteins such as Env, is important to characterize mRNA-LNP formulated antigens and their immune response. The application of the workflow should be of general interest to the vaccine field.

The following points need to be addressed to strengthen the conclusions of the manuscript:

Fig. 3: The SPR graphs should show the KD values. 3e shows gp151 ND binding to PGZL1; What is its affinity for gp151 MPER ND? Supp. Fig. 2 The KD measurements seem to be the result of one experiment? Could the authors provide triplicates for the MPER antibodies?

The authors describe three different SPR setups to test binding. Which ones were used for the data presented in 3c?

What were the affinities of the Ebola Gp antibodies?

The cryoEM structures are interesting as they define the MPER epitope to be more complex. The previously reported affinities of 10E8 to MPER peptides including partial or complete TM is approximately 10x higher. How do the authors explain this discrepancy? In fact, I would expect that the additional contacts of 10E8 with gp120 and gp41 would have increased the 10E8 binding affinity.

Does the removal of the glycosylation sites correlate with the additional 10E8 binding to gp120 and gp41 and thereby explain the increased 10E8 binding to gp151 MPER ND as compared to gp151 ND? The difference is 55-fold? In contrast, PGZL1 binding does not seem to be affected by the glycans? The angle of approach of different MPER bnAbs varies by up to approximately 34°; could all known MPER bnAbs bind to the structure of gp151 MPER ND? Can this be modelled?

Mode of binding: The authors should show a comparison of 10E8 binding to peptides and to the MPER epitope within the new structure. Are the MPER epitope contacts all the same?

Finally, are the additional contacts of 10E8 important for its function? Do mutations thereof affect neutralization?

Reviewer #2

(Remarks to the Author)

In this paper the authors describe an improved method for presenting glycoproteins in lipid nanodiscs; demonstrate three uses of such discs, for proteins from two different pathogens; and present beautiful structural data for an important HIV neutralizing antibody 10E8. This paper is a significant advance over the main authors' previous publication (Rantalainen, K.

et al. HIV-1 Envelope and MPER Antibody Structures in Lipid 915 Assemblies. Cell Rep 31, 107583 (2020)). The advances are: the standardization and higher throughput of the improved method, which should make it easier for other labs to adopt; and the higher resolution structural data.

This reviewer is enthusiastic about the paper. The method should be of great interest in structural biology and the 10E8 structure will be of interest in the HIV vaccine and antibody field. While this reviewer does not have expertise in some of the methods here, they seem quite complete. The figures are clear and visually appealing.

Major comments:

- The description of the comparison of the new 10E8 structure with the published peptide-only structure (PDB 4G6F), was a little hard to understand. A main or supplementary figure would help the reader visualize the comparison. This could include the full epitope-paratope regions of the paratope and epitope overlays separately.
- Authors should say something about the method optimization. What parameters did they test? Key for other labs that might use this method, what parameters were most important?
- What enabled the better EM structure? Are the discs substantially different from the ones described in the 2020 paper?
- Does the structure validate the predicted effects of the new mutations that the authors put into Env gp151 MPER ND? This should be analyzed.

Minor comments:

- These are sticky probes – hi background on naïve B cell population within splenocytes, which may limit utility. The authors see higher Ag+ in memory cells in 5/6 mice but do not show unvax mice, so formally the discs could be sticking nonspecifically to memory. It would be helpful to test on unvaccinated mice. This is what they did for NHP in fig 5. This reviewer suggests but does not require this experiment.
- Watch the order of supp figures – callout to S4 comes early
- Line 168 “higher” than what?
- Fig 4A legend: need to say what gray is
- Fig 4D legend: the mRNA did not encode nanodiscs, did it?
- Mention why you do flow using one probe in 2 colors
- line 342, “they will stabilize the trimer and prevent protomers from dissociating.” While you may expect that for an antibody like PGT145, this is unlikely to be an effect of VRC01 and BG18.
- Line 357 “Unlike in our previous WT Env AMC011 nanodisc 357 cryo-EM reconstruction, density for the TM domain remained undetected, presumably due to the R696S mutation disrupting the stabilizing connection point of the three helices in the middle of the bilayer.” Other hypotheses? Don’t presume.
- Make colors match in 6a and 6b (use 6b scheme in 6a)
- Line 389 “indicating that epitope presentation in a more native context recapitulates interactions more faithfully than epitope peptide alone” - we can’t know that, your new structure could have unknown artifacts. This reviewer would leave out that clause, and say “suggests” when making the same point in line 395.

Reviewer #3

(Remarks to the Author)

The manuscript by Rantalainen et al. reports a study seemingly proposing nanodisc-solubilized virus glycoprotein for vaccine candidate design. Although the title reads like that the authors are treating nanodisc approach as a vaccine platform, the paper itself only studies HIV envelope glycoprotein (Env), by mixing data from in vitro characterization with structural studies, but with little or no direct data to support that the nanodisc-solubilized Env would provide safe, effective vaccination in any primate animal model. The authors did present preliminary FACS data from mouse model and some data for using the nanodisc for sorting non-human primate cells. But unfortunately, these data insufficiently justify that the nanodisc-derived method would be a viable vaccine formula or platform. Using nanodisc to solubilize membrane proteins, including the HIV Env, have been extensively studied and documented in the literature. Without introducing any novel concept, this paper attempts to “rebrand” the nanodisc-based method as a vaccine design platform. Although this idea is hypothetically feasible, the current manuscript falls short in proving this idea. The major concerns are in the following.

- (1) What is the antigenicity and immunogenicity of the nanodisc itself in both mouse and primate models? Is it safe to use the present composition of the nanodisc to form an Env vaccine? What side effects are expected for these nanodiscs as a vaccine?
- (2) What is the difference of Env-solubilizing nanodisc particle as compared to mRNA-LNP in terms of lipid composition and immunogenicity?
- (3) Is the Env nanodisc designed to only elicit the MPER-targeting bnAbs? Can other epitopes of bnAbs be incorporated in the same design? Is the nanodisc necessary for presenting MPER-derived epitopes?
- (4) What is the advantage or disadvantage of the nanodisc Env as a vaccine candidate over those engineered soluble gp140 Envs?
- (5) Are there any cryo-EM densities of the nanodiscs in the antibody-bound Env? Are there any cryo-EM structural evidence

that the nanodiscs better preserve the bnAbs-eliciting MPER or any other epitopes?

(6) Are there any direct structural evidence that the Env nanodiscs are better vaccine design than gp140 Env mimetics?

(7) How is the MPER-targeting bnAbs modifying or causing the rearrangement of MPER structure? Would the presence of nanodisc make the MPER epitope less accessible for bnAbs? Have the nanodisc design been optimized to maximally expose the MPER epitopes? If so, how was it done?

(8) As the safety and side effects of mRNA-LNP still remains a concern for certain patient groups and demands in-depth investigation with more evidence, how do the authors recommend on the necessary steps for nanodisc to become successful as part of a vaccine design?

Version 1:

Reviewer comments:

Reviewer #1

(Remarks to the Author)

The authors responded to the previous concerns and included additional data that strengthen the manuscript for publication in Nature Communications

Reviewer #3

(Remarks to the Author)

The authors did not fully address my major concerns in terms of novelty and significance. Instead, the authors now admitted the previous proposal might be a bad one, and now just revised the phrase to re-sell the same results, without providing any further data to support fundamentally important advancement expected for this journal. Using nanodisc to study membrane proteins including Env has been around for decades, I don't see any major novelty and significance here. I am sorry that I cannot recommend its publication in Nature Communications.

We thank the reviewers for valuable comments. Please find point-by-point responses below

Reviewer #1 (Remarks to the Author):

The manuscript by Schief and colleagues describes the use of nanodiscs to generate soluble membrane anchored viral glycoproteins such as HIV-1 Env as tools for biophysical characterization of vaccine antigens that have been formulated as mRNA LNPs to produce membrane anchored immunogens. Furthermore, nanodisc-Env can be used to analyze antibody responses after vaccination. The authors employ the system to further analyze Env complex formation with the MPER-targeting bnAb 10E8.

Major findings are:

Env gp151 MPER ND with three gp41 glycosylation sites removed and a R696S mutation binds to 10E8 with nanomolar affinity. Removing the glycans helps to increase the affinity from 200 nM to 3.6 nM.

Glycosylation of Env gp151 ND is close to native Env glycosylation.

Biotinylated nanodisc-Env was used as a bait to sort B cells from mice immunized with Env gp151 MPER ND mRNA-LNPs. No MPER-specific responses were detected. Abs were cloned and confirmed no binding to MPER.

Nanodisc-Env can be used as tetramer probes to evaluate B cell responses in NHPs.

CryoEM structures of nanodisc-Env in complex with 10E8 Fabs showed 3:1 and 3:2 complexes, but no 3:3 complexes. Novel features of the 10E8 epitope include 10E8 contacts with the C-terminus of gp120 and with gp41 helices 6 to 8.

In summary, the described workflow to characterize membrane-anchored viral glycoproteins such as Env, is important to characterize mRNA-LNP formulated antigens and their immune response. The application of the workflow should be of general interest to the vaccine field.

Response: We thank the reviewer for the careful review and valuable comments

The following points need to be addressed to strengthen the conclusions of the manuscript:

Fig. 3: The SPR graphs should show the KD values. 3e shows gp151 ND binding to PGZL1; What is its affinity for gp151 MPER ND? Supp. Fig. 2 The KD measurements seem to be the result of one experiment? Could the authors provide triplicates for the MPER antibodies?

Response: We thank the reviewer for this suggestion. We added KD values to Figure 3. The previous Figure 3e exemplified a measurement with SPR modality C that, although highly sensitive, yields semi-quantitative binding results rather than monovalent \$K_D\$ s and therefore is an alternative approach for confirming antibody binding with high sensitivity and low sample consumption. To strengthen the conclusions and to report \$K_D\$ s, we performed additional SPR experiments with HIV and Ebola antibodies as Fabs (Modality A). These are now added to revised Fig. 3e and in Supp. Fig. 2, including a selection of MPER antibody affinities measured with modality A in triplicates. We also added additional text describing the differences between modalities.

Upon addressing this point, we noted a missing detail in the experiment description – all SPR experiments related to this point were done with nanodiscs assembled with DOPC lipids only. The purpose was to simplify the baseline experiments by excluding the possible lipid headgroup

charge effect to MPER antibody binding. We have corrected the materials and methods section and figure legends accordingly.

The authors describe three different SPR setups to test binding. Which ones were used for the data presented in 3c?

Response: Figure 3c uses modality A (nanodisc capture – Flow Fab). This is now indicated in the figure legend and we have indicated this also in the figure next to SPR data

What were the affinities of the Ebola Gp antibodies?

Response: We have now expressed Ebola antibodies EBOV-296 and EBOV-13C6 as Fabs and measured the affinities to EBOV GP Mayinga ND (46 nM and 16 nM, respectively). These are now given in Fig. 3 e.

The cryoEM structures are interesting as they define the MPER epitope to be more complex. The previously reported affinities of 10E8 to MPER peptides including partial or complete TM is approximately 10x higher. How do the authors explain this discrepancy? In fact, I would expect that the additional contacts of 10E8 with gp120 and gp41 would have increased the 10E8 binding affinity.

Does the removal of the glycosylation sites correlate with the additional 10E8 binding to gp120 and gp41 and thereby explain the increased 10E8 binding to gp151 MPER ND as compared to gp151 ND? The difference is 55-fold? In contrast, PGZL1 binding does not seem to be affected by the glycans?

Response: We thank the reviewer for the insightful comment. We propose that the previously reported higher affinity to partial MPER peptide is stemming from the absence of steric restrictions imposed by additional gp120 and gp41 glycans and lipid surface contacts. This is supported by increased affinity of 10E8 to Env gp151 MPER ND where three glycans around the MPER epitope are deleted. We have now included this in the discussion with relevant references.

The angle of approach of different MPER bnAbs varies by up to approximately 34°; could all known MPER bnAbs bind to the structure of gp151 MPER ND? Can this be modelled?

Mode of binding: The authors should show a comparison of 10E8 binding to peptides and to the MPER epitope within the new structure. Are the MPER epitope contacts all the same?

Response: We have added an additional supplementary figure 5 to illustrate these important comparisons. Aligning published crystal structures with MPER peptide in our cryo-EM structure allows visualizing how the binding pose in crystal structures would be accommodated in the context of the entire Env and the lipid bilayer. This analysis shows that these crystal structures may represent a binding mode adopted in the absence of complete epitope and that MPER antibodies can adopt a wide range of binding modes. We also noted that only 14 out of 29 contacting residues in the paratope and 15 out of 26 contacting residues in the epitope were observed in the 10E8 crystal structure compared to the cryo-EM structure. We have included these observations in results and discussion.

Finally, are the additional contacts of 10E8 important for its function? Do mutations thereof affect neutralization?

Response: We thank the reviewer for this important question. We have now conducted neutralization assays with key point mutations discovered from the new structure. The results are presented in Fig 7. These data show that reverting paratope residues of 10E8 FRH3 to their

germline sequences results in up to eightfold loss in neutralization efficiency with 10E8. Modest loss in neutralization efficiency was observed with several epitope mutations with notable exception of L660A resulting in approx. sevenfold increase. These observations are now included in the last paragraph of results section and in discussion. Together they show that newly identified contacts can influence 10E8 neutralization and be used to improve vaccine designs.

Reviewer #2 (Remarks to the Author):

In this paper the authors describe an improved method for presenting glycoproteins in lipid nanodiscs; demonstrate three uses of such discs, for proteins from two different pathogens; and present beautiful structural data for an important HIV neutralizing antibody 10E8. This paper is a significant advance over the main authors' previous publication (Rantalainen, K. et al. HIV-1 Envelope and MPER Antibody Structures in Lipid Assemblies. Cell Rep 31, 107583 (2020)). The advances are: the standardization and higher throughput of the improved method, which should make it easier for other labs to adopt; and the higher resolution structural data.

This reviewer is enthusiastic about the paper. The method should be of great interest in structural biology and the 10E8 structure will be of interest in the HIV vaccine and antibody field. While this reviewer does not have expertise in some of the methods here, they seem quite complete. The figures are clear and visually appealing.

Response: We thank for the positive and appreciative review of the manuscript.

Major comments:

- The description of the comparison of the new 10E8 structure with the published peptide-only structure (PDB 4G6F), was a little hard to understand. A main or supplementary figure would help the reader visualize the comparison. This could include the full epitope-paratope regions of the paratope and epitope overlays separately.

We removed Supp. Fig panel 4e and extended this analysis in Supplemental Fig 5 to address similar point from Reviewer #1. While the contacts within the MPER peptide remained the same between the two structures, the crystal structure included only 14 out of 29 contacting residues in the paratope and 15 out of 26 contacting residues in the epitope when compared to the cryo-EM structure.

- Authors should say something about the method optimization. What parameters did they test? Key for other labs that might use this method, what parameters were most important?

Response: We agree that this is a critical point and have added a sentence in methods section emphasizing the importance of the batch assembly approach where glycoprotein is bound to affinity matrix during nanodisc assembly. While we initially tested a variety of GP:scaffold:lipid ratios, they did not have a significant difference in the nanodisc assembly efficiency when batch assembly approach was used. Batch assembly also improved the reproducibility of the entire method which is critically important for scalable routine use of the approach.

- What enabled the better EM structure? Are the discs substantially different from the ones described in the 2020 paper?

Response: This very important point is now addressed in Methods section. We attribute the improved resolution of the EM structure mainly to stable Env-Fab complexes and higher protein concentration achieved through improved nanodisc assembly conditions. This allowed us to use a fluorinated Fos-choline detergent in the EM grid preparation. This detergent is known to improve

orientation distribution of particles by blocking the air-water interface and does not disturb the lipid bilayer could only be used at high nanodisc protein concentrations. Additionally, rapid progress of cryo-EM instrumentation and data collection allowed us to test more conditions and collect larger datasets.

- Does the structure validate the predicted effects of the new mutations that the authors put into Env gp151 MPER ND? This should be analyzed.

Response: Mutations intended to improve 10E8 binding were glycan knockouts (KOs). As expected, we did not observe densities for these glycans. In these parts the absence of features in the EM structure confirmed the intended effect of the mutations together with the increased affinity in SPR measurements. Although we were unable to resolve the transmembrane domain, the lack of density in the middle of the nanodisc indicates that the R696S mutation has disrupted the stabilizing interaction between the three TM helices. In our earlier work using WT Env construct, we were able to observe strong density, albeit at low resolution connecting the three helices in the middle of the bilayer (Rantalainen, K. et al. HIV-1 Envelope and MPER Antibody Structures in Lipid Assemblies. Cell Rep 31, 107583 (2020)).

Minor comments:

- These are sticky probes – hi background on naïve B cell population within splenocytes, which may limit utility. The authors see higher Ag+ in memory cells in 5/6 mice but do not show unvax mice, so formally the discs could be sticking nonspecifically to memory. It would be helpful to test on unvaccinated mice. This is what they did for NHP in fig 5. This reviewer suggests but does not require this experiment.

Response: We included this important control to our early method optimization where we detected no binding to probes using unimmunized C57BL/6 (B6) mouse splenocytes (Fig 4a). We agree that this is an important control and should be included in future pre-clinical assessments. We can also monitor interactions with lipid membranes by including an empty nanodisc control in SPR when measuring apparent affinities of monoclonal antibodies from sorted B cells to immunogens in nanodiscs (Supplementary figure 3b).

- Watch the order of supp figures – callout to S4 comes early

Response: We thank the reviewer. We have moved the reference to the EM control experiment to a more appropriate location (Results: GP nanodiscs in cryo-EM structural studies) and revised figure callouts.

- Line 168 “higher” than what?

Response: Added “compared to soluble Env gp140”

- Fig 4A legend: need to say what gray is

Response: Reference to MSP1D1 scaffold negative control was added

- Fig 4D legend: the mRNA did not encode nanodiscs, did it?

Response: Nanodiscs with Env sequences matching the immunogen were used for SPR. We have now corrected and clarified the figure legend.

- Mention why you do flow using one probe in 2 colors

Response: Two colors were used to enrich for double-positive cells that are binding the antigen and not one of the fluorophores. This is now clarified in the first paragraph of FACS results.

- line 342, “they will stabilize the trimer and prevent protomers from dissociating.” While you may expect that for an antibody like PGT145, this is unlikely to be an effect of VRC01 and BG18.

Response: We agree and have rephrased the sentence. The stabilization may rather be a secondary effect of antibodies preventing the trimer from adhering to the air-water interface, which could prevent the trimer from falling apart due to freeze denaturation upon EM grid preparation.

- Line 357 “Unlike in our previous WT Env AMC011 nanodisc 357 cryo-EM reconstruction, density for the TM domain remained undetected, presumably due to the R696S mutation disrupting the stabilizing connection point of the three helices in the middle of the bilayer.” Other hypotheses? Don’t presume.

Response: We have rephrased this. It could also be possible that lack of intracellular C-terminal domain could make the TM domain connection less stable.

- Make colors match in 6a and 6b (use 6b scheme in 6a)

Response: We thank the reviewer for noting this and have matched the colors in 6a and b.

- Line 389 “indicating that epitope presentation in a more native context recapitulates interactions more faithfully than epitope peptide alone” - we can’t know that, your new structure could have unknown artifacts. This reviewer would leave out that clause, and say “suggests” when making the same point in line 395.

Response: We have changed this to “suggest”. While the nanodisc is a considerably more native context for the MPER epitope compared to MPER peptide alone, as supported by the well-defined ectodomain binding pocket, we may have introduced other unknown artifacts as the reviewer correctly pointed out.

Reviewer #3 (Remarks to the Author):

The manuscript by Rantalainen et al. reports a study seemingly proposing nanodisc-solubilized virus glycoprotein for vaccine candidate design. Although the title reads like that the authors are treating nanodisc approach as a vaccine platform, the paper itself only studies HIV envelope glycoprotein (Env), by mixing data from in vitro characterization with structural studies, but with little or no direct data to support that the nanodisc-solubilized Env would provide safe, effective vaccination in any primate animal model. The authors did present preliminary FACS data from mouse model and some data for using the nanodisc for sorting non-human primate cells. But unfortunately, these data insufficiently justify that the nanodisc-derived method would be a viable vaccine formula or platform. Using nanodisc to solubilize membrane proteins, including the HIV Env, have been extensively studied and documented in the literature. Without introducing any novel concept, this paper attempts to “rebrand” the nanodisc-based method as a vaccine design platform. Although this idea is hypothetically feasible, the current manuscript falls short in proving this idea. The major concerns are in the following.

Response: We would like to apologize for any confusion. The manuscript does not intend to propose to use nanodiscs as vaccines. Rather it provides an analytical platform to support vaccine development. We have changed the title to **Virus glycoprotein nanodisc platform for vaccine**

analytics to better reflect this important distinction. As we indicated in the abstract (line 34), mRNA-LNP can now deliver transmembrane glycoprotein vaccine candidates, but biophysical and structural characterization still relies on recombinant protein production and soluble mimics are often used in place of immunogen sequences that encode the transmembrane protein. We see mRNA-LNP as a superior delivery mechanism for transmembrane immunogens (e.g. covid19 vaccines or, for HIV applications, see Parks et al. SciTransMed 2025, Ramezani-Rad et al. SciTransMed 2025, Xie Z. et al. Science 2025, Melzi et al. Immunity 2022). Therefore we did not attempt to produce recombinant protein/nanodisc-based immunogens. Some of the reasons include poor translatability of nanodisc assembly to clinical material production, including endotoxin free production of scaffold protein and generally low expression levels of transmembrane recombinant proteins. Instead, we present a platform that allows the critically important in vitro characterization methods of the immunogens and analysis of immune responses which, to the best of our knowledge, have not been described previously. These include novel concepts for 1) accurate affinity measurement with SPR at high throughput using transmembrane glycoproteins in nanodiscs and 2) sorting of B cells with nanodisc probes that contain identical sequences to mRNAs that were used for immunization. In addition we present the most complete and detailed epitope of an MPER targeting HIV antibody with a novel binding pocket targeted by 10E8, facilitated by methodological advances in nanodisc preparation for cryoEM. This structure may be critically important for MPER targeted vaccine development as the structural template for rational design of this epitope component has thus far been missing.

(1) What is the antigenicity and immunogenicity of the nanodisc itself in both mouse and primate models? Is it safe to use the present composition of the nanodisc to form an Env vaccine? What side effects are expected for these nanodiscs as a vaccine?

Response: We believe mRNA-LNP is far better immunogen delivery platform compared to recombinant protein nanodisc glycoproteins and did not intend to present nanodiscs as an immunogen delivery modality.

(2) What is the difference of Env-solubilizing nanodisc particle as compared to mRNA-LNP in terms of lipid composition and immunogenicity?

Response: Currently the lipid composition that is associated with in vivo-expressed Env from mRNA-LNPs is unknown but assumed to be related to the lipid composition of the cells that are transfected in vivo. Deciphering this would be a major undertaking and is outside the scope of this study.

(3) Is the Env nanodisc designed to only elicit the MPER-targeting bnAbs? Can other epitopes of bnAbs be incorporated in the same design? Is the nanodisc necessary for presenting MPER-derived epitopes?

Our structure (Fig 6) indicates that the N332 glycan supersite targeted by bnAbs like BG18, and the CD4-binding site targeted by bnAbs like VRC01 and N6, can both be engaged simultaneously with the MPER site. We see that assembling designs that integrate multiple engineered epitopes into nanodiscs could allow testing the responses more broadly by biophysical and structural means, while the nanodiscs themselves may not be needed (and are not currently intended) as immunogens. The MPER epitope may be presented alone as a peptide on liposomes or partially in soluble Env trimer designs, but our structural characterization of the complete epitope indicates that these may not recapitulate the entire epitope and can thus lead to impaired immune

responses compared to transmembrane Env designs such as BG505 MD39.3 gp151 that has performed well in NHPs (Ramezani-Rad et al. SciTransMed 2025) and humans (Parks et al. SciTransMed 2025).

(4) What is the advantage or disadvantage of the nanodisc Env as a vaccine candidate over those engineered soluble gp140 Envs?

As noted in earlier responses, using nanodiscs as immunogens would bring significant disadvantages as they would be challenging to produce as endotoxin free material in adequate amounts and could lead to off-target responses to scaffold proteins. Engineered soluble gp140 immunogens are generally truncated before the MPER epitope and therefore do not generally present the MPER. mRNA can deliver engineered transmembrane Env designs in a native-like context and we thus consider them as the most promising approach to present a complete set of bnAb epitopes including the MPER.

(5) Are there any cryo-EM densities of the nanodiscs in the antibody-bound Env? Are there any cryo-EM structural evidence that the nanodiscs better preserve the bnAbs-eliciting MPER or any other epitopes?

We can visualize the nanodisc portion of the sample by low-pass filtering the cryo-EM data as indicated in Fig 6a. While only nanodiscs can preserve the entire MPER -targeting bnAb epitope (lipid surface, MPER peptide and the rest of the ectodomain), we don't have evidence indicating that other epitopes would be differently engaged in nanodisc assembled Env vs. soluble Envs.

(6) Are there any direct structural evidence that the Env nanodiscs are better vaccine design than gp140 Env mimetics?

Env nanodiscs allow studying transmembrane Env designs using approaches that have traditionally been available only for soluble mimetics such as gp140. We don't propose using Env nanodiscs as vaccines. Nanodiscs do allow structural studies of the MPER epitope that is deleted in gp140 mimetics and therefore we consider the structure in Fig 6 to be evidence that nanodiscs are suitable for structural studies of MPER targeting antibodies, unlike gp140 mimetics.

(7) How is the MPER-targeting bnAbs modifying or causing the rearrangement of MPER structure? Would the presence of nanodisc make the MPER epitope less accessible for bnAbs? Have the nanodisc design been optimized to maximally expose the MPER epitopes? If so, how was it done?

We designed the Env gp151 MPER ND construct to improve the MPER epitope accessibility by removing glycans that could mask the epitope and by adding the point mutation R696S in the TM domain. The point mutation was discovered in a directed evolution experiment as described in methods section *Identification of R696S mutation on HIV Env by mammalian directed evolution*.

(8) As the safety and side effects of mRNA-LNP still remains a concern for certain patient groups and demands in-depth investigation with more evidence, how do the authors recommend on the necessary steps for nanodisc to become successful as part of a vaccine design?

Although nanodiscs are not envisioned as a practical platform for preparation of clinical trial material, they can provide valuable additional data points for mRNA-LNP investigations e.g. by examining how antibodies from mRNA-LNP immunizations engage the transmembrane Env in the membrane context. They could also be used in combination with mRNA formulations in early

pre-clinical experiments to assess transmembrane designs as protein administered vaccines vs. mRNA-LNPs, although we have not pursued this later avenue.